# Downward cloud venting of the Central African biomass burning plume during the West Africa summer monsoon

Alima Dajuma[1,3], Kehinde O. Ogunjobi[1,4], Heike Vogel[2], Peter Knippertz[2], Siélé Silué[5],

Evelyne Touré N'Datchoh[3], Véronique Yoboué[3], Bernhard Vogel[2]

[1] West African Science Service Center on climate change and Adapted land Use (WASCAL)/Federal University of Technology Akure, Ondo state, Nigeria
[2] Institute of Meteorology and Climate Research, Karlsruhe Institute of Technology (KIT)
[3] University Félix Houphouët Boigny Abidjan, Côte d'Ivoire
[4] Federal University of Technology Akure (FUTA), Ondo state, Nigeria
[5] University Péléforo-Gbon-Coulibaly, Korhogo, Côte d'Ivoire

Correspondence to: Alima Dajuma (alima.dajuma@yahoo.com)

**Abstract.** Between June and September large amounts of biomass burning aerosol are released into the
atmosphere from agricultural fires in Central and southern Africa. Recent studies have suggested that this plume is carried westward over the Atlantic Ocean at altitudes between 2 and 4 km and then northward with the monsoon flow at low levels to increase the atmospheric aerosol load over coastal cities in southern West Africa (SWA), thereby exacerbating air pollution problems. However, the processes by which these fire emissions are transported into the planetary boundary layer (PBL) are still unclear. One potential factor is the large-scale
subsidence related to the southern branch of the monsoon Hadley cell over the tropical Atlantic. Here we use convection-permitting model simulations with COSMO-ART to investigate for the first time the contribution of downward mixing induced by clouds, a process we refer to as downward cloud venting in contrast to the more common process of upward transport from a polluted PBL. Based on a monthly climatology, model simulations compare satisfactory with wind fields from reanalysis data, cloud observations, and satellite retrieved carbon
monoxide (CO) mixing ratio. For a case study on 02 July 2016, modelled clouds and rainfall show overall good agreement with Spinning Enhanced Visible and InfraRed Imager (SEVIRI) cloud products and Global Precipitation Measurement Integrated Multi-satellite Retrievals (GPM-IMERG) rainfall estimates. However, there is a tendency for the model to produce too much clouds and rainfall over the Gulf of Guinea. Using the CO dispersion as an indicator for the biomass burning plume, we identify individual mixing events south of the
coast of Côte d'Ivoire due to midlevel convective clouds injecting parts of the biomass burning plume into the PBL. Idealized tracer experiments suggest that around 15% of the CO mass from the 2–4km layer are mixed to below 1km within two days over the Gulf of Guinea and that the magnitude of the cloud venting is modulated by the underlying sea surface temperatures. There is even stronger vertical mixing when the biomass burning plume reaches land due to daytime heating and a deeper PBL. In that case, the long-range transported biomass
burning plume is mixed with local anthropogenic emissions. Future work should provide more robust statistics on the downward cloud venting effect over the Gulf of Guinea and include aspects of aerosol deposition.

**1 Introduction**

The interest in air pollution over southern West Africa (SWA) has grown substantially in recent years (Knippertz et al., 2015). Population growth, urbanization, and industrialization have led to increasing local emissions from various sources in addition to natural ones. For instance, coastal city development in SWA is leading to more traffic and fuel consumption (Doumbia et al., 2018). Anthropogenic emissions are expected to continue increasing if no regulations are implemented (Liousse et al., 2014). Domestic fires, traffic, and waste burning are the most important sources of pollution in West Africa (Marais and Wiedinmyer, 2016; Bahino et al., 2018; Djossou et al., 2018). In the framework of the Dynamics-Aerosol-Chemistry-Cloud Interactions in West Africa (DACCIWA; Knippertz et al., 2015a) field campaign in SWA during June-July 2016 (Flamant et al., 2018b), measurements of the French Service des Avions Français Instrumentés pour la Recherche en Environnement (SAFIRE) ATR-42 aircraft showed fairly high background concentrations (i.e., outside of urban plumes) with PM1 mass concentrations dominated by secondary organic compounds that contribute 53% to the total aerosol mass (Brito et al., 2018). For the urban pollution plumes of Abidjan, Accra, and Lomé, they found a doubling of PM1 mass concentrations. Air pollution is a main concern for human health leading to respiratory and other diseases (Lelieveld et al., 2015) but it may also affect local meteorology. For instance, using model sensitivity experiments Deetz et al. (2018) showed that increasing aerosol loadings can lead to a reduced inland penetration of the Gulf of Guinea Maritime Inflow (Adler et al., 2019) and a weakening of the nocturnal low-level jet over SWA.

During the summer West African monsoon (WAM), the atmosphere over SWA is characterized by a mixture of pollutants from different sources as highlighted by Knippertz et al. (2017). In addition to the local pollution, long-range transport of dust from the Sahel and the Sahara Desert as well as biomass burning aerosol from Central and southern Africa affect the atmospheric composition. Mineral dust has been shown to affect radiation, precipitation, and many WAM features (e.g., Konare et al., 2008; Solmon et al., 2008; Stanelle et al., 2010; Raji et al., 2017; N'Datchoh et al., 2018). During this period, biomass burning is widespread in Central and southern Africa, when plumes are carried westward by a jet between 2 and 4 km (Barbosa et al., 1999; Mari et al., 2008), while in West Africa activity peaks during the dry season from October to March (N'Datchoh et al., 2015). Biomass burning is an important source of aerosols and trace gases, with an estimated combined emission of several thousand Tg a$^{-1}$ for tropical areas (Barbosa et al., 1999; van der Werf et al., 2003; Ito and Penner, 2004). For instance the estimated carbon emissions from both tropical fires and fuel wood use was 2.6 Pg C a$^{-1}$ over the period 1998–2001 (van der Werf et al., 2003). Hao and Liu (1994) estimated that almost half of tropical biomass burning emissions come from Africa, with savanna fires contributing up to 30% to the global total and 64% to the African total. During the DACCIWA field campaign, a surprisingly high level of pollution was observed over the sea upstream of SWA. Haslett et al. (2019) found a significant mass of aged accumulation mode aerosol in the planetary boundary layer (PBL) over both continent and ocean. According to modelling work by Menut et al. (2018) biomass burning from Central and southern Africa increases the level of air pollution in urban cities such as Lagos and Abidjan by approximately 150 μg m$^{-3}$ for carbon monoxide (CO), 10–20 μg m$^{-3}$ for ozone (O$_3$), and 5 μg m$^{-3}$ for particulate matter with diameters smaller than 2.5 μm (PM2.5).

An important and open question is the mechanism by which the biomass burning aerosols from Central Africa get from the layer of midlevel easterlies into the PBL. Das et al. (2017) reported that global aerosol models tend

to simulate a quick descent to lower levels just off the western coast of Africa, while Cloud-Aerosol Lidar with Orthogonal Polarization (CALIOP) observations suggest that smoke plumes continue their horizontal transport at elevated levels above the marine boundary layer. The strength and speed of subsidence vary amongst models and subregions. The hypothesis we investigate in this paper is that clouds play a considerable role in the downward mixing of biomass burning aerosol from the elevated plume. Most previous studies have focused on cloud-induced upward transport of aerosols and chemical species from close to their sources in the PBL to the free troposphere (e.g., Dickerson et al., 1987; Ching et al., 1988; Cotton et al., 1995). Using two-dimensional idealized simulations, Flossmann and Wobrock (1996) calculated the mass transport of trace gases across cloud boundaries and from the marine boundary layer into the free troposphere. The atmospheric condition was a well-mixed boundary layer up to about 500m and above a moist layer of about 2.2 km capped by a very dry stable layer. They found that 60% of the mass of an inert tracer is exported due to convective clouds. The same paper also examines the transport of $SO_2$ including chemical reactions showing that only a small fraction is dissolved and reacts in the aqueous phase, while substantial amounts of $SO_2$ are redistributed by clouds (see also Kreidenweis et al., 1997). Using a 1-D entraining-detraining plume model with ice microphysics, Mari et al. (2000) studied the transport of CO (inert tracer), $CH_3COOH$, $CH_2O$, $H_2O_2$, and $HNO_3$, and compared the results with observations from the Trace and Atmospheric Chemistry Near the Equator-Atlantic (TRACE-A) campaign. Convective enhancement factors at 7–12 km altitude, representing the ratios of post convective to pre-convective mixing ratios, were calculated for both observed and simulated cases. Observed (simulated) values were 2.4 (1.9) for CO, 11 (9.5) for $CH_3COOH$, 2.9 (3.1) for $CH_2O$, 1.9 (1.2) for $H_2O_2$, and 0.8 (0.4) for $HNO_3$. Pickering et al. (1996) showed an upward transport of CO mixing ratios, $NO_x$ and hydrocarbons by convective clouds during the Brazilian phase of the TRACE-A experiment. Moreover, Yin et al. (2001) simulated trace gas redistribution by precipitating continental convective clouds and found abundant highly soluble gases in their uppermost parts, while Halland et al. (2009) showed substantial vertical transport of tropospheric CO by deep mesoscale convective systems.

In contrast, rather few studies investigated the downward transport of elevated pollution through convective clouds. For the marine PBL, aerosol particles from the free troposphere have been identified to serve as cloud condensation nuclei in stratiform clouds with cloud entrainment contributing up to 20% of the aerosol mass (Raes, 1995; Katoshevski et al. 1999). Over land, most studies concentrated on the Amazon rainforest. Based on campaign data during the wet season, Betts et al. (2002) showed that convective downdrafts rapidly transport air with high ozone down to the surface from around 800 hPa, suggesting a significant role of this process for the photochemistry of the PBL and surface ozone deposition. Gerken et al. (2016) even found evidence for transport of ozone-rich air from the mid-troposphere to the surface, enhancing the volume mixing ratio in the boundary layer by as much as 25 ppbv on the regional scale, while Wang et al. (2016) demonstrated the injection of high concentrations of small aerosol particles into the PBL by strong convective downdrafts. In more general terms, Jonker et al. (2008) proposed a refined view of mass transport by cumulus convection relevant for the dispersion of aerosol. According to them, the descending motion near the cloud environment is significant and rather different than in a distant cloud environment, which is characterized by more uniform and quiescent dry descending motion.

This study uses simulations with the COSMO (Consortium for Small-Scale Modelling) model (Baldauf et al., 2011) online coupled with Aerosol and Reactive Trace gases (ART; Vogel et al., 2009) to investigate cloud-induced transports of biomass burning aerosols from mid-level tropospheric layers into the PBL over the Gulf of Guinea and SWA. A one-month simulation for July 2016 (i.e., during the DACCIWA field campaign) over a large domain will be evaluated with available observational datasets and combined with a detailed high-

resolution case study, followed by idealized tracer experiments designed to quantify the vertical transport. The paper is organized as follows. Section 2 describes the satellite and re-analysis data as well as the model framework and simulation set up used for this study. The model evaluation is presented in section 3. In section 4 the downward cloud venting process and its contribution to the vertical mixing of the biomass burning plume are assessed and discussed. Analysis of an artificial tracer to quantify the mass fraction of the biomass burning

plume that mixes down into the PBL is given in section 5. The last section presents a summary of the results and conclusions.

## 2 Data and Modelling

### 2.1 Observational Data

The following data from space-borne platforms and reanalysis are used for this study:

1. The Moderate Resolution Imaging Spectroradiometer (MODIS) is a key instrument on board the Earth Observing System (EOS) Terra satellites. The instrument views the entire Earth's surface every one to two days acquiring data in 36 spectral bands ranging in wavelengths from 0.4 μm to 14.4 μm. The MODIS product for cloud properties we use is MOD08_E3 (Platnick et al., 2017). Amongst others it contains 1x1 degree grid

averaged values of cloud fraction averaged over the month of July 2016.

2. Measurement of Pollution in the Troposphere satellite (MOPITT) is used to derive CO volume mixing ratios. MOPITT measurements are performed in eight nadir-viewing spectral channels using a gas correlation spectroscopy technique with a horizontal resolution of $22x22$ km$^2$ (Clerbaux et al., 2008b). A detailed description of the instrument and measurement technique can be found in Drummond and Mand (1996), Pan et

al. (1998), and Edwards et al. (1999). The data are available at different height levels from the surface to 150 hPa. Global coverage is reached after 3 to 4 days. MOPITT data have been shown to distinguish CO pollution from large cities and urban areas from background pollution using only thermal infrared information (Clerbaux et al., 2008) and to perform even better using a combination of thermal infrared and solar radiation in the PBL (Buchwitz et al., 2007; Turquety et al., 2008). Kar et al. (2008) highlighted that retrievals in the lower

troposphere over continental areas provide reasonable information on surface emissions of CO, although the measurements suffer from strong thermal contrasts. According to Buchholz et al. (2017), MOPITT measurements overestimate relative to ground-based remote sensing Fourier transform infrared spectrometer with a bias of less than 10% evaluated over 14 stations.

3. To represent standard meteorological fields, monthly mean ERA-Interim reanalysis data from the European

Centre for Medium-Range Weather Forecasts (ECMWF) at a spatial resolution of 0.25° are used for this study (Dee et al., 2011).

4. Daily Sea Surface Temperatures (SST) from the National Oceanic and Atmospheric Administration (NOAA; Reynolds et al., 2007) are analyzed for the detailed case study on 02 July 2016. The SST analysis has a spatial resolution of 0.25 degrees and a temporal resolution of one day. The product uses Advanced Very High Resolution Radiometer (AVHRR) satellite data from the Pathfinder AVHRR SST dataset (Stowe et al., 2002).

5. The Global Precipitation Measurement Integrated Multi-satellite Retrievals (GPM-IMERG) product from National Aeronautics and Space Administration (NASA) is used for rainfall evaluation. It uses an algorithm that merges precipitation radar, microwave precipitation estimates, microwave-calibrated infrared, and rain gauge analyses at a spatial resolution of 0.1° over the latitudinal belt 60°N–60°S. The product has a time resolution of 30 minutes (Hou et al., 2014; Huffman et al., 2018).

All observational data (satellites and reanalysis data) are collocated with respect to time and space for the comparison with the model results.

## 2.2 Modelling

For the simulations performed in this study, the numerical weather prediction model of the Consortium for Small-Scale Modelling (COSMO; Baldauf et al., 2011) online coupled with Aerosol and Reactive Trace gases (ART) is used (Vogel et al., 2009). COSMO-ART allows the treatment of aerosol dynamics, atmospheric chemistry, and the feedback with radiation and cloud microphysics (Vogel et al., 2009; Knote et al., 2011; Bangert et al., 2012; Athanasopoulou et al., 2013). A 1 D-plume rise model of biomass burning aerosols and gases in COSMO-ART calculates online the injection height of the biomass pollution plume and the emission strength of gases and particles (Walter et al., 2016). The parameterization scheme uses data obtained from the Global Fire Assimilation System (GFAS v1.2; Kaiser et al., 2012), in particular MODIS satellite data of the fire radiative power. Anthropogenic emission data are taken from the Emission Database for Global Atmospheric Research Hemispheric Transport of Air Pollution version 2 (EDGAR HTAP_v2; Edgar, 2010) for 2010 with a 0.1° horizontal resolution. In addition, the recently developed gas flaring emission parameterization for SWA by Deetz and Vogel (2016) was used, which is based on a combination of remote sensing observations and physically based combustion equations. Biogenic emissions, sea salt, dimethyl sulfide, and mineral dust are calculated online within the model system. Meteorological initial and boundary conditions are taken from operational global ICOsahedral Non-hydrostatic (ICON) model (Zängl et al., 2015) runs of the German Weather Service (DWD). Initial and boundary conditions for gaseous and particulate compounds are derived from forecasts using the Model for Ozone and Related Chemical Tracers (MOZART; Emmons et al., 2009).

In order to cover a large domain including the fire areas in Central Africa and, at the same time, to reach a high horizontal resolution in our area of interest (Gulf of Guinea and SWA), we used the nesting option of COSMO-ART. The modelling domains are presented in Fig. 1. The outer domain D1 (18°W–26.6°E; 20°S–24.6°N) indicated by the small box at the bottom left corner of the figure covers West and western Central Africa as well as the adjacent southeastern Atlantic Ocean. The red rectangle inside this box shows the location of the nested domain D2 (9°W–1°E; 3°–10.8°N), mostly covering Ivory Coast and Ghana. The color shading gives the surface height above sea level. D2 is dominated by tropical forests in the south to savanna and grassland vegetation in the north. Simulations are run on D1 with a horizontal grid spacing of 5 km and 50 verticals levels.

The simulation over D2 is nested into D1 with a horizontal grid spacing of 2.5 km with 80 verticals levels up to
30 km (28 levels below 1.5 km above sea level).

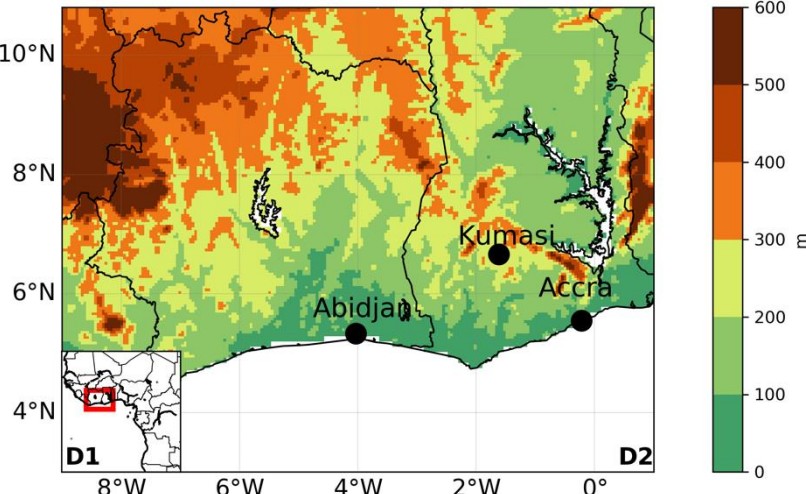

*Figure 1. Geographical overview. Model domains D1 (inset) and D2 (main image). The horizontal resolution in case of D1 is 5 km and 2.5 km in case of D2. The color shading gives the surface height above sea level over D2. The black dots mark the largest cities of the region Abidjan, Accra and Kumasi.*

The model configuration used in this study is the same as in Deetz et al. (2018). Both domains, D1 and D2, were run with the parametrization for deep convection switched off and using the two-moment microphysics scheme (Seifert and Beheng, 2006). Over D1, the modelled period ranges from 25 June–31 July 2016 with the meteorological state being re-initialized every day at 00 UTC. ICON operational forecasts at 13 km grid spacing with 90 vertical levels are used as meteorological initial and boundary conditions and MOZART chemistry with a grid mesh of 280 km x 213 km and 56 vertical levels for the pollutant initial and boundary data. Cloud condensation nuclei are prescribed with a constant aerosol number concentration of 1700 cm$^{-3}$. The purpose of the D1 simulation is to compare the model output and observations for monthly mean conditions, i.e., for July 2016, after a six-day spin-up.

In addition, we analyze a particular case study on 02–03 July 2016 simulated over D2 using the outputs of D1 for both meteorological and chemical initial and boundary conditions. The period 02–03 July 2016 was chosen because it falls into the post onset phase of the monsoon, characterizing an undisturbed monsoon condition, and is thus favorable for process studies (Knippertz et al., 2017; Deetz et al., 2018). The two-moment microphysics scheme was combined with the prognostic aerosol, this way accounting for aerosol direct and indirect interactions. The purpose of this run is to perform detailed process studies, in particular with respect to the cloud-induced mixing over the Gulf of Guinea. An artificial tracer experiment is performed to quantify the percentage of mass mixed from the free troposphere into the PBL. We use CO as an inert tracer and a surrogate for biomass burning emissions. The deposition velocity is set to zero and chemistry switched off in order to account only for meteorological atmospheric transport processes. Interactions between gas phase chemistry, aerosol dynamics, and meteorology are neglected. We set a constant profile of 1 ppmv at the height where the maximum concentration of the biomass burning plume is observed (i.e., 2–4 km) and 0 below and above that

layer. This concentration is held constant at the domain boundaries during integration, while mixing processes can change it in the interior.

## 3 Model evaluation

Figure 2a shows a July 2016 average of the wind speed and streamlines at 925 hPa as simulated by COSMO-ART. Figure 2b shows the corresponding figure for the ERA-Interim re-analysis. The wind is southeasterly in the southern hemisphere and turns southwesterly along the Guinea Coast after crossing the equator. This low-level monsoon flow advects relatively cool and moist air from the Gulf of Guinea onto the continent. In July the precipitation maximum is located around 10°N (e.g., Janicot et al., 2008), and westerlies penetrate far north into

the continent and over the adjacent Atlantic Ocean. Apart from a slightly northward shifted turning point and more fine-scale detail in the higher resolved COSMO-ART data, the agreement with ERA-Interim in terms of the overall structure of the low-level flow field is good. However, there are some prominent differences in wind speed. ERA-Interim shows highest wind speeds in the southern hemisphere and a slow down towards SWA as well as a clear minimum over Central Africa. COSMO-ART simulates a stronger monsoon flow and also

significantly higher winds over Central Africa. Maxima reach 15 m s$^{-1}$ in both model and reanalysis. COSMO-ART shows a domain average of 6 m s$^{-1}$, 1.4 m s$^{-1}$ higher than ERA-Interim.

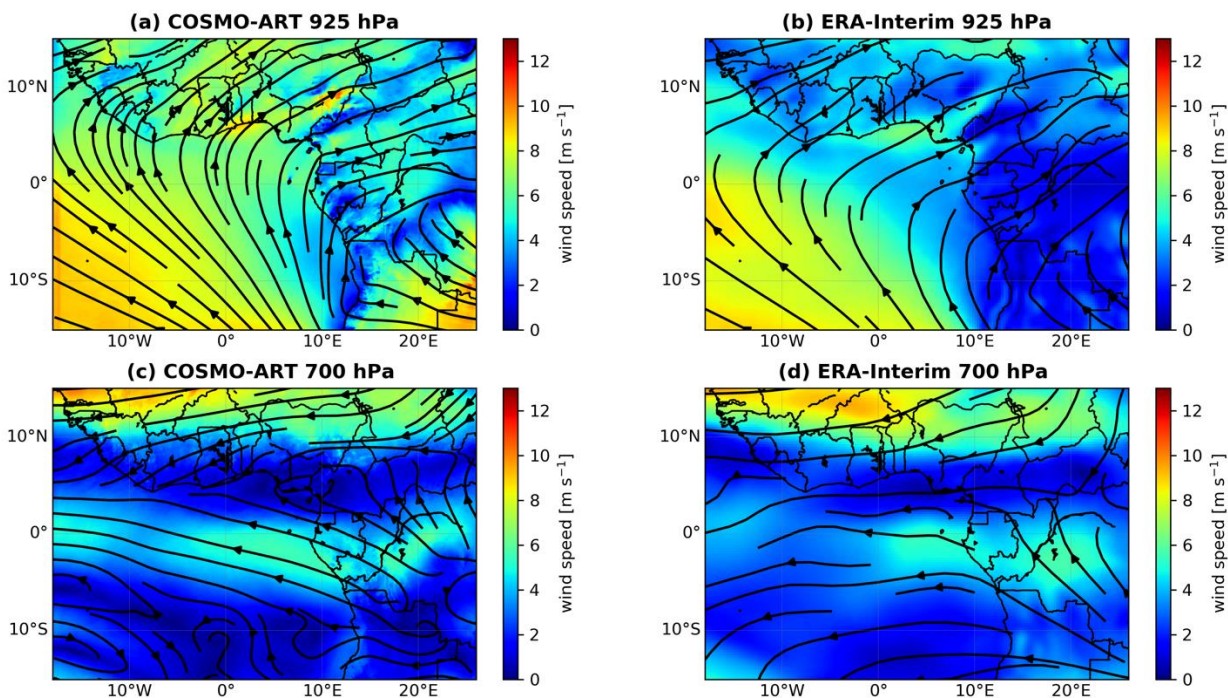

*Figure 2. Average wind speed (color shading) and streamlines at 925hPa (top) and 700hPa (bottom) simulated with COSMO-ART (left) and in ERA-Interim re-analysis (right) for July 2016. Note the different scales in top*

*and bottom panels.*

The wind field at 700 hPa is characterized by a broad easterly flow across most of the considered domain (Figs. 2c and d). A maximum is found over the Sahel known as the African Easterly Jet (AEJ), which typically peaks

around 600 hPa (Parker et al., 2005) and is the result of the large meridional temperature gradient at low levels (Cook, 1999; Wu et al., 2009). The AEJ is well represented in COSMO-ART with a maximum wind speed of 10.4 m s$^{-1}$ as compared to 9.73. m s$^{-1}$ in ERA-Interim. Easterlies are also enhanced near the equator, to the south of an area with weaker flow over the Guinea Coast. There are some subtle differences between COSMO-ART and ERA-Interim here, with the former showing a larger northward component over the ocean and slightly stronger winds. COSMO-ART also displays more fine structure in the southern hemisphere, where winds are overall weaker. Despite the moderate differences discussed above, we anticipate an overall realistic transport of biomass burning aerosol in the model, i.e., westward away from the hotspots in Central Africa out to the Atlantic and then northward into SWA with the monsoon flow, if downward mixing occurs.

The simulated total cloud fraction averaged over July 2016 (Fig. 3a) is compared to observations from MODIS (Fig. 3b). SWA is very cloudy in summer with typical values ranging from 70% to almost 100% in agreement with a multi-year climatology presented in Hill et al. (2016). The cloud cover is overall adequately represented by COSMO-ART over land, particularly relative to the poor performance of many coarser-resolution climate models (Hannak et al., 2017). Cloud cover maxima stretch from southwestern Ghana to northeastern Ivory Coast, along the Atakora chain at the border of Ghana and Togo, and over the Guinea Highlands of Liberia and Sierra Leone with overall satisfactory agreement between the two datasets. Towards the Sahel, to the north of 8ºN, cloud fraction decreases in COSMO-ART but much less so in MODIS, which only shows a prominent minimum over central Ivory Coast. Over the Gulf of Guinea, cloud cover is clearly overestimated by the model, suggesting a potential overestimation of cloud-induced mixing. The two local minima upwind of Abidjan and Accra may be related to coastal upwelling but are hard to verify with MODIS due to the coarser resolution.

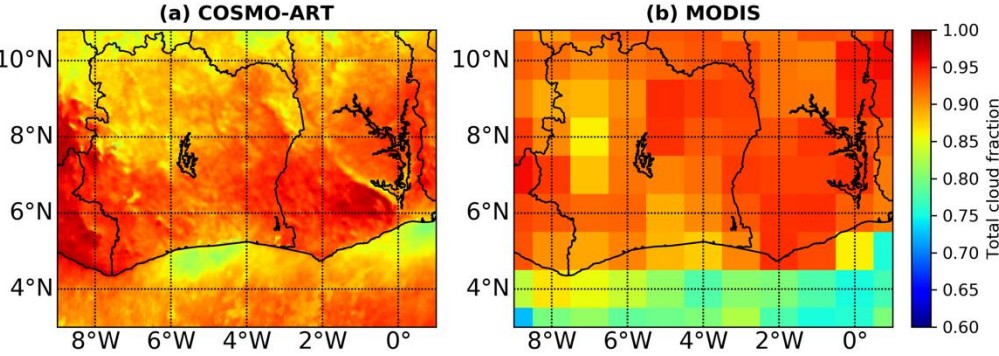

*Figure 3. Monthly mean total cloud fraction for July 2016 over domain D2 as simulated with COSMO-ART (a) and observed by MODIS (b).*

Finally, the modelled mixing ratio of surface CO (Fig. 4a) is evaluated with satellite data from MOPITT for July 2016 (Fig. 4b). A gridded monthly mean of CO from MOPITT is computed using the daily mean CO retrieved for the 1000–900 hPa layer. Some areas have too frequent cloud contamination and therefore do not allow the computation of a representative monthly mean (white shading in Fig. 4b). Overall the spatial patterns of CO concentration are captured by the model with some discrepancies. Over Central Africa widespread burning is evident with a larger magnitude and spatial extent in the model as compared to MOPITT. From there, a plume of enhanced concentrations stretches northwestward in both datasets, but again values in COSMO-ART are

somewhat larger and therefore reach more remote parts of the Atlantic Ocean. This also supports a potential overestimation of the import of pollution from Central Africa into SWA in the model. In addition, COSMO-ART simulates marked pollution plumes over Nigeria associated with Lagos, the oil fields in the Niger Delta (flaring activities), and the Sahelian city of Kano, which are hard to verify with MOPITT due to cloud contamination. However, emissions from Kano, where clouds are less frequent than in the south, are likely overestimated. Emissions from other large cities (e.g., Accra, Kumasi, Abidjan) in contrast appear relatively weak in COSMO-ART. This may be at least partly due to uncertainties in standard emission inventories (Liousse et al., 2014). Despite some overall discrepancies, we argue that the two fields are similar enough to draw conclusions on the importance of cloud-mixing process in the model, particularly because the fields are relatively similar over the ocean.

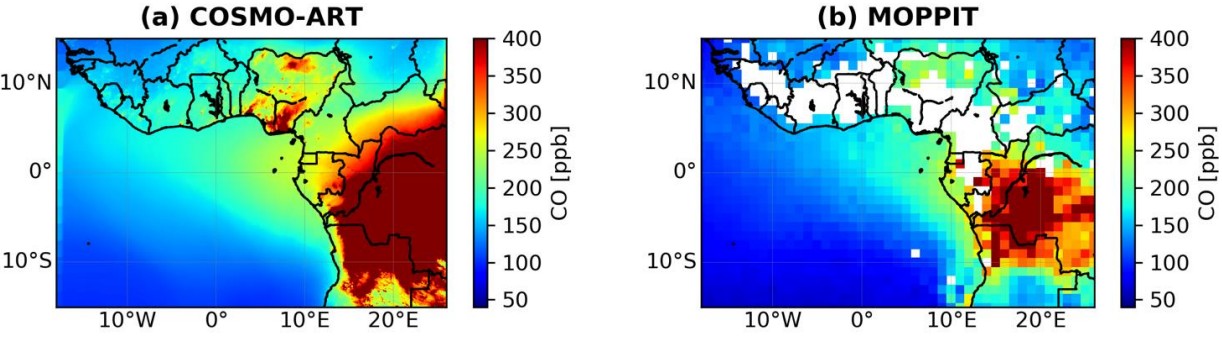

*Figure 4. Monthly mean surface CO concentrations for July 2016 as simulated by COSMO-ART (left) and as observed by MOPPIT (right).*

## 4 A Case study

In section 3 we presented simulated monthly mean conditions. We will now focus on a case study for 02 July 2016 to illustrate the impact of meteorology on the spatial and temporal distribution of CO. We will especially focus on the role of convective clouds on the vertical distribution of CO.

### 4.1 Simulated temperature distribution

The spatial distribution of simulated 2m temperature at 12 UTC on 02 July 2016 is displayed in Fig. 5. At this time of day the temperature is already higher over land than over the ocean. Local temperature maxima are located over cities such as Abidjan and Accra. High temperatures are also simulated in the central part of Ghana near Lake Volta. Modelled temperatures over the Gulf of Guinea are between 26°C and 28 °C in agreement with observed SSTs shown in Knippertz et al. (2017). In Fig. 5 there are clear indications of cold pools related to convective cells developing over the Gulf over Guinea and the adjacent land areas, particularly over southern Ivory Coast (see Fig. 6c for precipitation). The hourly analysis of the temperature field (not shown) shows cold

pools first appearing around 07 UTC and persisting during the whole day. They are connected to downward motion starting at and above cloud base, bringing air and its constituents from aloft into the PBL.

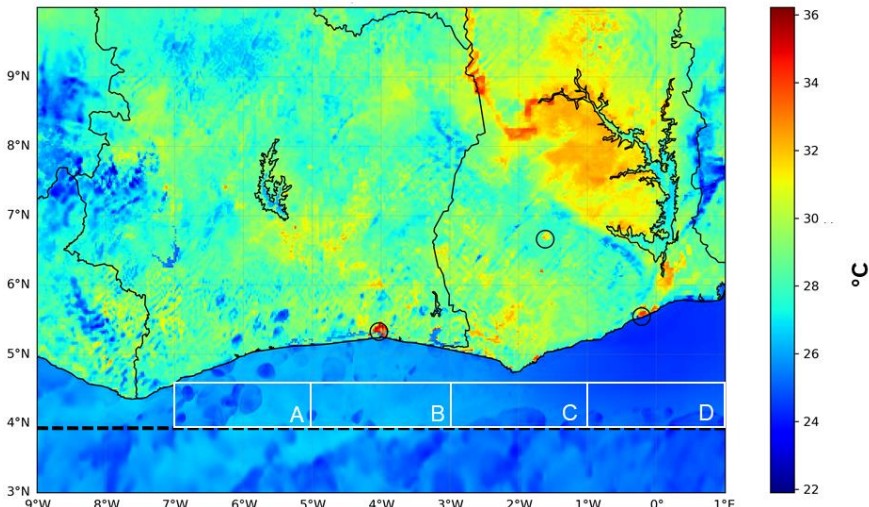

*Figure 5. 2-m temperature as simulated by COSMOS-ART over domain D2 (see Fig. 1) on 02 July 2016 at 12 UTC. Note the small-scale cold-pool signatures over the ocean. The circles mark the urban heat islands of*
*Abidjan, Kumasi and Accra. The position of the zonal cross-section in Fig. 8 is marked with a dashed line. Different subdomains are defined between 4–4.7°N that will be used for the tracer experiments discussed in section 5 (see Fig. 11): 7–5°W (A), 5–3°W (B), 3–1°W (C), and 1°W–1°E (D).*

**4.2 Spatial distribution of clouds and rainfall**

Satellite retrieved images from EUMETSAT on 02 July 2016 show widespread clouds over SWA and the adjacent ocean, with convective cells located over the Gulf of Guinea south of Ivory Coast at 12 UTC (Fig. 6b). They produce rain rates of several mm h$^{-1}$ in the course of the afternoon according to GPM-IMERG (Fig. 6d). The cells over the ocean developed near the border between Ivory Coast and Ghana in the morning hours and propagated slowly westward in the course of the day (not shown). They formed despite anomalously cold
coastal waters but may have benefitted from substantially warmer SSTs nearer the equator (see Fig. 3 in Knippertz et al., 2017). Mostly moderate precipitation is also observed over land, in central Ghana, around Kumasi as well as along the borders between Cote d'Ivoire with Liberia, Guinea, and Mali.

Corresponding total cloud cover and precipitation as simulated by COSMO-ART are shown in Figs. 6a and c. In the model the whole area is dominated by clouds (Fig. 6a) with moderate gaps around Lake Volta and over the
ocean upwind of Ghana and Ivory Coast. There is reasonable qualitative agreement between the model and observations (Fig. 6b) but the differences in cloud optical thickness evident from the satellite image make a detailed comparison somewhat difficult. With respect to precipitation, COSMO-ART shows substantially more fine structure than GPM-IMERG. Many localized showers are evident over Ivory Coast and neighboring countries with higher intensities over the hilly terrain in Liberia and along the land-sea breeze convergence
parallel to the coast. Larger cells form in the model over the hills surrounding Lake Volta. The largest and most

intense convective systems are simulated over the ocean with a pronounced north–south elongation along the southwesterly monsoon flow. These were persistent throughout the day (not shown). Despite the differences in resolution there is overall good qualitative agreement between model and observations, in particular with respect to the maxima over central Ghana and Liberia. Convection in the north is underestimated and convection over

the ocean is overestimated by COSMO-ART in agreement with the cloud biases evident from Fig. 3. The latter further confirms that cloud-induced mixing may be somewhat overestimated by COSMO in this specific case, allowing only a rather qualitative assessment.

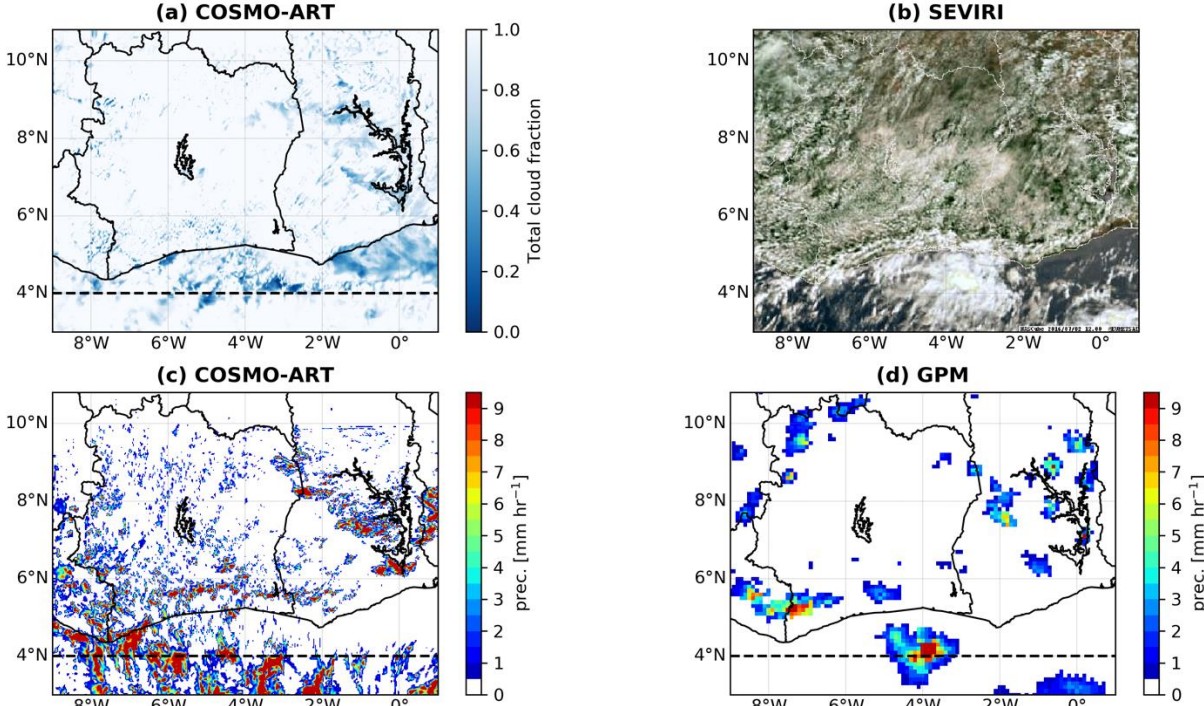

*Figure 6. Spatial distribution of clouds and rainfall over domain D2 (see Fig. 1) on 02 July 2016. (a) Total*
*cloud fraction simulated by COSMO-ART at 12 UTC. (b) Spinning Enhanced Visible and InfraRed Imager (SEVIRI) cloud visible image from EUMETSAT at 12 UTC (from http://nascube.univ-lille1.fr). (c) Precipitation rate simulated by COSMO-ART at 18 UTC and (d) corresponding fields from the GPM–IMERG rainfall estimates. The position of the zonal cross-section in Fig. 8 is marked with dashed lines.*

### 4.3 Simulated and observed spatial distribution of CO

Figure 7 presents the simulated spatial distribution of the CO concentration for 02 July 2016, 12 UTC at about 500m and 2000m above the ground over the domains D1 (Figs. 7a and b) and D2 (Figs. 7c and d). At 500 m (Fig. 7a) there is a stark concentration difference between land and ocean with thick pollution plumes over the biomass burning areas in Central Africa (Barbosa et al., 1999; Mari et al, 2008; Zuidema et al., 2016) and over

Nigeria. The urban plumes from coastal cities such as Abidjan, Cotonou, Lomé, and Lagos are also visible. These results come from the high anthropogenic emissions used in our study, which have maxima over Nigeria and the big cities along the coast. The simulated hourly CO concentrations (not shown here) reveal that there is

a north-eastward transport of CO from the local sources in the PBL with the southwesterly monsoon flow (Knippertz et al., 2017; Deroubaix et al., 2018). However, Flamant et al. (2018a) also showed that parts of the urban pollution can re-circulate to the near-coastal waters after being mixed into the midlevel easterly or sometimes northeasterly flow. Significantly lower but still considerable CO concentrations are simulated in the marine PBL over the entire eastern tropical Atlantic including the Gulf of Guinea. There is a local enhancement next to the coast stretching from Cameroon to Ivory Coast. At this height level, CO is transported with the south-westerly monsoon winds from the ocean toward SWA coastal cities (see Fig. 2). Compared to the monthly mean concentration of CO (Fig. 4), 02 July was characterized by elevated pollution levels, especially over Nigeria. Concentrations over the nested domain D2 at 500m (Fig 7c) are moderated with traces of higher CO concentrations over the Gulf of Guinea, some smaller elongated plumes (e.g., from Abidjan and Accra), and much elevated levels downstream of Lake Volta. As concentrations above ground level are shown in Fig. 7c, the elevated values over the Atakora chain at the border of Ghana with Togo are at least party related to the fact that higher ground is closer in the vertical to the main midlevel pollution plume from Central Africa.

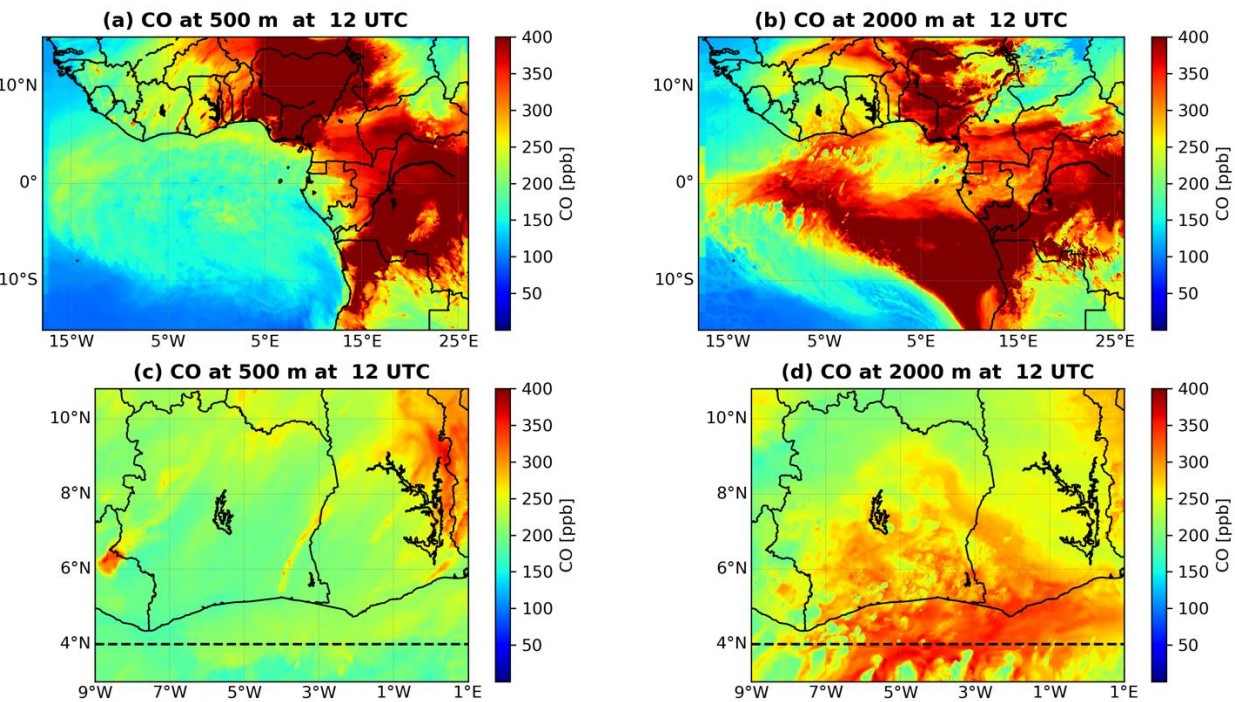

*Figure 7. CO concentrations on 02 July 2016 at 12 UTC as simulated by COSMO-ART at (a) ca. 500 m and (b) ca. 2000 m above ground level over D1, and at (c) ca. 500 m and (d) ca. 2000 m over D2.*

Aloft at approximately 2000 m (Fig. 7b) the CO distribution is fundamentally different. Maximum CO concentrations with values of about 400 ppbv are found over the eastern Atlantic Ocean, downstream of the main burning areas in southern and Central Africa. High concentrations stretch far into SWA (e.g., into Burkina Faso, Mali, and Niger), even in areas where 500 m concentrations are not that high, as for example over Ivory Coast (Fig. 7a). This clearly suggests a relationship to long-range transport of biomass burning plumes and is further corroborated by much reduced values in a simulation where biomass burning emissions are suppressed (not shown). The elevated concentrations at 500 m over the ocean to the west and north of the main plume at

2000 m (i.e. over the equatorial Atlantic Ocean near 15°W and arching into the Gulf of Guinea) suggest downward mixing into the PBL from aloft, which is further elucidated in the following paragraph. Subsidence within the high pressure system west of the African continent may also support the downward transport of CO into the PBL (Zuidema et al., 2016). Zooming in on domain D2 (Fig. 7d), concentrations at 2000 m are generally much higher than at 500 m (Fig. 7c), in particular over the coastal zone. Strikingly some marked "holes" are evident that correspond to areas of convective cells and associated cold pools (see Figs. 5 and 6c), suggesting that in these areas clouds support downward mixing.

To further investigate this hypothesis, vertical distributions of CO concentrations and cloud liquid water content from model output are considered (Fig. 8). High CO concentrations are simulated over most of the Gulf of Guinea but levels are generally higher between 10°W and 0° (upwind of Cote d'Ivoire and Ghana) than between 0° E to 5°E (Fig. 7b) and we will therefore concentrate on this region.

Figures 8a and b show zonal cross sections of CO over the Gulf of Guinea at 4°N (i.e., close to the coastal cities of SWA) over D2 on 02 July 2016 at 12 UTC and 18 UTC, respectively. There is a clear band of high CO concentrations of up to 400 ppbv, mostly between 1 and 3.5km over D2, which is the signature of the long-range transport of the biomass burning plume from Central Africa (Mari et al., 2008; Zuidema et al., 2016), possibly affected by larger-scale subsidence. Several stripes of low concentration are simulated and these structures become more pronounced at 18 UTC (Fig. 8b). They are related to simulated (and also observed) convective clouds (Fig. 6) that transport CO into the PBL from above. Analyzing the simulated diurnal cycle (not shown). we found that over the ocean clouds appear after 07 UTC and are persistent throughout the day, while CO becomes increasingly visible in the PBL and eventually reaches the surface. Concentrations below 1km can reach up to 60% of the maximum located at midlevel height due to downward mixing.

Figures 8c and d show meridional-vertical cross-sections of, respectively, CO concentration and specific cloud liquid water content along 6°W, close to where convective activity is seen in Fig. 8a. Areas of high cloud liquid water are collocated with minima in CO, supporting the idea of cloud-induced transport and mixing. The most prominent of such areas is located around 4.3°N, where significant amounts of cloud water stretch from below 500m to almost the top of the biomass burning plume, leading to an substantial erosion.

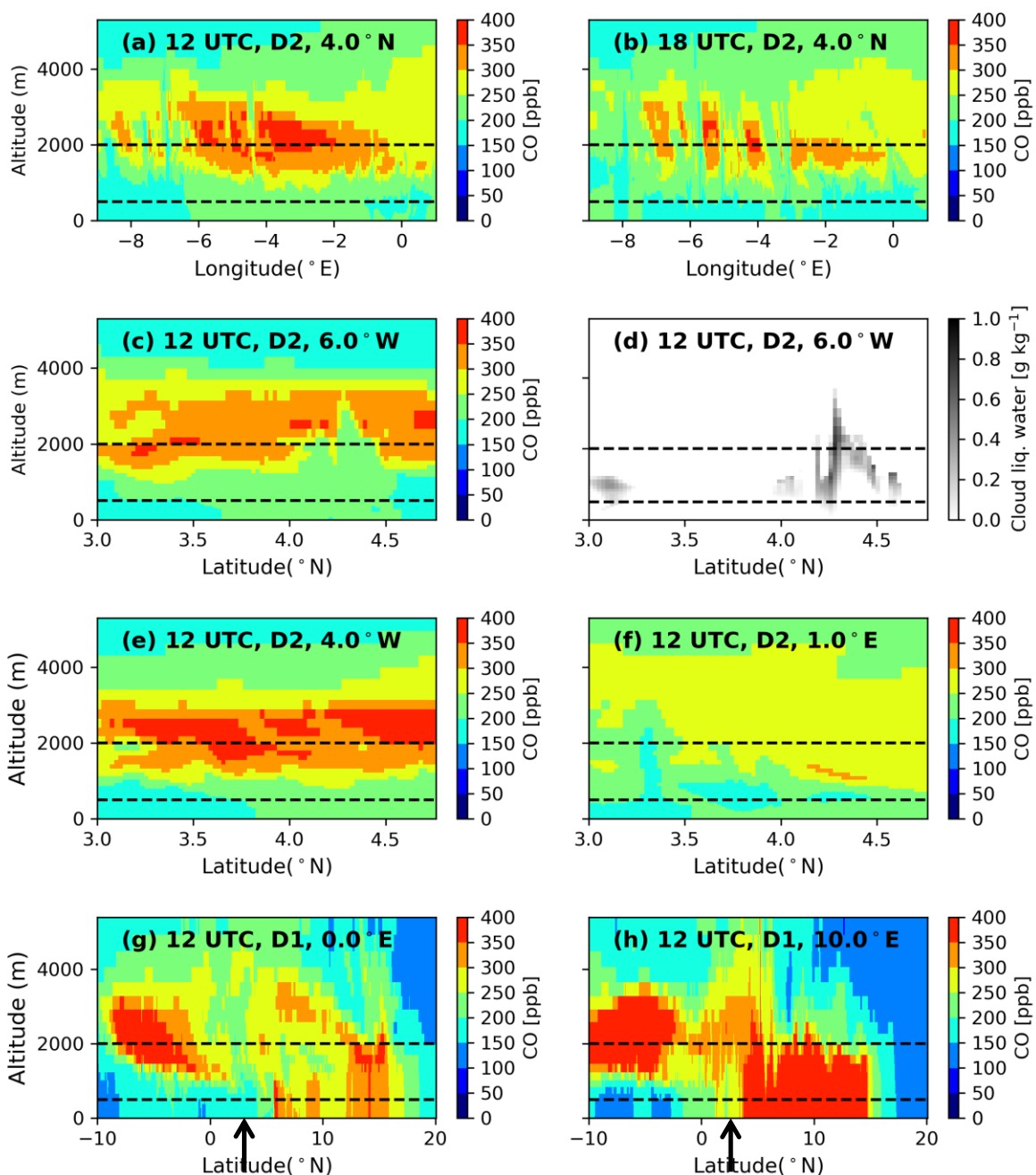

*Figure 8. Vertical cross-sections of CO concentration and cloud liquid water content as simulated by COSMO-ART over domains D1 and D2 (see Fig. 1). Zonal-vertical cross-sections at 4°N of the CO concentration at (a) 12 UTC and (b) 18 UTC on 02 July 2016 for domain D2. (c) Meridional-vertical cross-section at 12 UTC at 6°W and (d) corresponding cloud liquid water content, both for domain D2. (e) and (f) As (c) but for 4°W and 1°E, respectively. (g) and (h) Meridional vertical cross-sections over D1 at 0°E and 10°E, respectively. Heights are given in altitude above ground level and the levels shown in Fig. 7 are marked by dashed lines in all panels. The arrows in (g) and (h) indicate the coast line.*

For comparison we also analyze meridional cross-sections of CO corresponding to Fig. 8c but at longitudes of 4°W and 1°E over domain D2. At 4°W (Fig. 8e) there are no pronounced gaps in the pollution plume, suggesting less convective mixing at this time than at 6°W but concentrations at low levels are not much different. There is even a slight increase northwards that may come from turbulent mixing or zonal advection into the section. In contrast to that, the cross-section at 1°E (Fig. 8f), which corresponds to the ocean offshore of the border of Ghana with Togo, shows a much weaker biomass burning plume in agreement with Fig. 7b. This appears to be the result of the bulk of the plume travelling westward over the ocean before turning northward into SWA. Also here, a slow decent of the lower boundary of the plume is visible. This may come from large-scale subsidence associated with the southern branch of the Hadley cell and/or from turbulent mixing.

Finally, it is also interesting to place the mixing near SWA into the larger regional context. Meridional cross sections along 0° and 10°E but reaching from 10°S to 20°N illustrate the full complexity of the plume evolution. At 0°E (Fig. 8g) there is a distinct biomass burning plume centered at 5°S. The skewed shape of this feature suggests a relatively fast northward transport around 1000m above ground level. Individual mixing events are evident (green spikes underneath the main plume in Fig. 8g). North of the coast (marked by an arrow in Fig. 8g) there is a complicated vertical structure with local near-surface emissions, overhead advection, and vertical mixing to various degrees, particular during the daytime shown here. Farther to the east at 10°E (Fig. 8h) the situation bears some similarities but the local emissions from Nigeria appear to play a larger role over land and the lofted biomass burning plume is more prominent over the immediate coast (see also Fig. 7b).

**5 Downward cloud venting**

The discussion in section 4 suggests that long-range transported biomass burning aerosol from Central Africa can be mixed into the PBL over the Gulf of Guinea in association with convective clouds. We will refer to this process as downward cloud venting in the following in contrast to the more classical upward cloud venting (e.g., Cotton et al., 1995).

In general, processes that can support the transport of biomass burning aerosols from free-tropospheric layers into the PBL include: (i) large-scale subsidence and thus vertical advection (Katoshevski et al., 1999), (ii) turbulent mixing through the PBL top, and (iii) vertical transport associated with convective clouds. With respect to point (i) we can state that the cross sections in Fig. 8 do not show clear indications of a systematic sinking of the biomass burning plume, suggesting that for the situation presented in section 4 synoptic-scale subsidence is not a leading factor.

To investigate the relative importance of processes (ii) and (iii), we designed an idealized tracer experiment. For the simulations starting at 02 July 2016 at 00 UTC initial profiles of a tracer were prescribed within the domains D1 and D2. The idealized tracer has a concentration of 1 ppmv between 2 and 4 km and is zero elsewhere. Chemical reactions as well as dry deposition are neglected in order to isolate effects of transport. At the lateral boundaries the tracer concentrations were held constant at the initial profile such that only mixing within the domain can change tracer concentrations. Two types of simulations were done: one with and one without

turbulent diffusion of the tracer. The idea behind this is to separate this effect from that of downward cloud venting. The simulations were carried out for a period of two days (2–3 July 2016).

For the larger domain D1, Figure 9 shows the percentage of tracer mass located between 2 and 4 km, between 1 and km, and below 1 km, the latter two with and without turbulent tracer diffusion. As the PBL is usually quite shallow over the ocean (as indicated for example by the low cloud base in Fig. 8d), the lowest layer should in most cases comprise the entire PBL and possibly also the lower part of the free troposphere with some variations in space and time. All values are averaged between 9°W and 1°E and the different panels show time
evolutions along different latitude circles.

Over the open ocean at 5° and 0°S (Figs. 9a, b), where mostly shallow cumuli are present, the concentration in the layer between 2 and 4 km stays fairly high over the 2-day period with well over 80% still present at the end of the simulation. The increase of the tracer mass in the intermediate layer from 1–2 km starts almost immediately after the begin of the simulation and reaches a plateau well above 20% during day 2. This indicates
that this increase cannot solely be the result of vertical mixing near the shown latitude circle but must also be related to horizontal transport of tracer that was mixed downwards upstream. The final tracer amount is almost independent of whether turbulent diffusion is considered or not, indicating the importance of downward cloud venting (compare dashed and solid red lines in Figs. 9a and b). The tracer reaches the layer below 1 km at the end of the second day parallel to a marked increase in the layer above. The tracer mass then slowly increases to
reach final values ranging around 15%. Here, the vertical mixing clearly is a combination of turbulent diffusion and cloud-induced mixing with the former contributing on the order of one quarter to one third. It is interesting to note that the southernmost section has a higher percentage of cloud-induced mixing than that over the equatorial cold tongue. Below we will show evidence that the higher SST in the former region likely supports cloud formation and associated mixing.

Over the Gulf of Guinea between 4 and 4.7°N (i.e., the latitude range used for Fig. 10) there is a more pronounced decrease of the tracer mass in the layer between 2 and 4 km down to about 65% after 2 days (Figure 9c). Consistently, the 1–2 km layer gains more tracer mass and exceeds 30% on the second day with a more continuous rise than farther south (Figs. 9a and b). Turbulent diffusion appears to play a more important role here, but the overall contribution is still fairly small. The increase of mass below 1 km, however, does not match
these differences, reaching values similar to that over the open tropical Atlantic but with a similar contribution from cloud induced mixing of about 60%. This result illustrates the complicated configuration of differential advection at different height levels combined with spatially differing vertical transport.

To investigate the aspect of SST influence further, Fig. 10 shows the final mixing state in the layer below 1 km after two days of integration, i.e., the right-hand-side intercept of the black curves in Fig. 9 but for steps of 1°
latitude. Plotted against SSTs for the same longitudinal range, a close correspondence is evident in the southern hemisphere. Both mixing and SSTs have a marked maximum around 6–7°S followed by a common minimum around 2°S. To the north, SSTs increase to levels even higher than in the southern hemisphere but the tracer mass increases only little. One likely reason for this is the smaller tracer concentrations aloft as evident from Fig. 9c. Another factor maybe the flow of near-surface air over the cooler equatorial water leading to a decrease

in buoyancy. It is also possible that enhanced shallow subsidence closer to the coast (see Fig. 8f) helps suppress
vertical mixing into the PBL.

Finally over land, i.e., between 5 and 10°N, the vertical exchange maximizes leading to a reduction in the 2–4
km layer down to almost 50% (Fig. 9d). Consistently, tracer mass in the intermediate layer increases more
strongly up to well over 44%, while tracer mass below 1 km reaches 23%. A clear diurnal cycle is evident,
particularly in the lower layer, with vertical mixing mostly occurring in the early afternoon when the PBL is
deepest. The suppressing of turbulent diffusion reduces the tracer mass by 20% with some evidence of a diurnal
cycle in the differences evident from Fig. 9d. As expected, dry mixing is more important in the early afternoon,
while cloud-induced mixing peaks later. The important role of clouds in vertical mixing over land is consistent
with the large cloud cover shown in Fig. 3. The diurnal cycle is also evident at 1–2km, where switching-off
turbulent diffusion leads to a net increase in this layer during the afternoon.

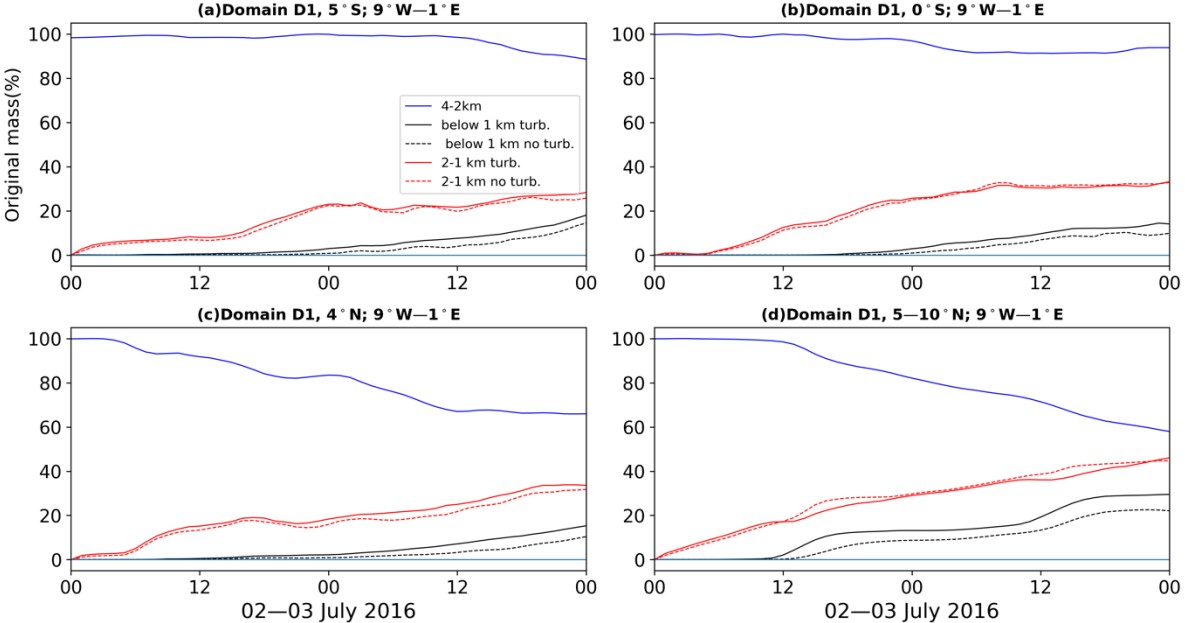

*Figure 9. Time evolution of the idealized tracer experiment on 02 and 03 July 2016 over Domain D1. Shown are changes in original mass in % between 2 and 4km (blue), between 1 and 2km (red) and below 1km (black). For the latter results including turbulent diffusion are shown by the solid lines and those without by the dashed*
*lines. Fields are averaged from 9°W–1°E along (a) 5°S (southeastern Atlantic), (b) 0° (equatorial cold tongue), (c) 4–4.7°N (Gulf of Guinea), and (d) 5–10°N (inland SWA).*

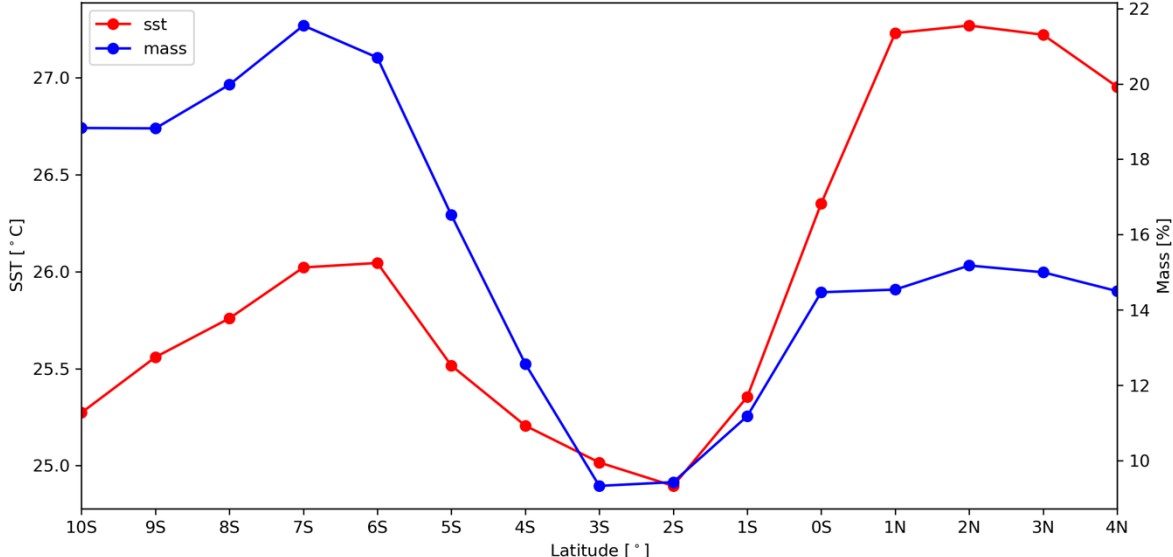

*Figure 10. Relationship between sea surface temperatures (SST) and vertical mixing of CO. CO masses in % (in blue) correspond to the values for the below 1-km layer at the end of the time window shown in Fig. 11 but for steps of one degree latitude. SSTs (in red) are from the Advanced Very High-Resolution Radiometer (AVHRR) and were averaged in the same way as the tracer concentration field.*

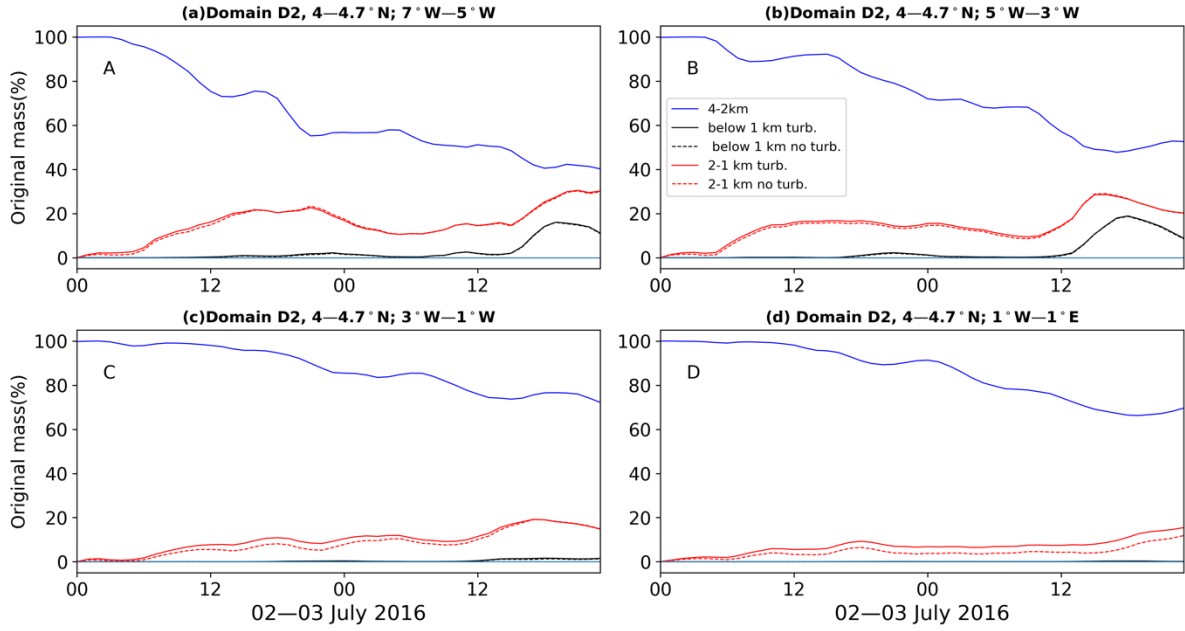

*Figure 11. Time evolution of the idealized tracer experiment on 02 and 03 July 2016 over Domain D2. Shown are changes in original mass in % between 2 and 4km (blue), between 1 and 2km (red), and below 1km (black). For the latter two results including turbulent diffusion are shown by the solid lines and those without by the dashed lines. Fields are averaged from 4°–4.7°N along (A) 7°–5°W, (B) 5°–3°W, (C) 3°–1°W, and (D) 1°W–1°E.*

Figure 11 shows results analogous to Fig. 9 but for the model domain D2 and for the areas A–D indicated in Fig. 5. Their latitudinal extent corresponds to Fig. 9c but the longitudinal extent varies between the different panels. Taken together the four panels of Fig. 11 stretch from 7°W to 1°E, while Fig. 9c stretches from 9°W to 1°E. Zooming down into such small areas illustrates the impact of short-lived intense local mixing events. While all four subregions show a marked decline in tracer mass in the 2–4km layer (blue lines), the evolution is sometimes bumpy and final values range from 41% in Fig. 11a to about 74% in Figs. 11c and d. Mixing into the layer immediately below the biomass burning plume (i.e., 1–2km, red lines) begins a few hours after the start of the simulations with even a transient reduction in some areas, mostly during morning hours. It is possible that the diurnal cycle in monsoon flow and cloudiness over the near landmass contributes to such fluctuations. Some correspondence is seen between "loss" events in the upper-layer and "gain" events in the middle layer but the connection is not always clear-cut, indicating effects of horizontal transport and possibly also mixing upwards from the main aerosol layer. As the 1–2km layer is above the PBL in most cases, it is no surprise that switching off turbulent diffusion has very little affect, apart from the easternmost domain, where clouds appear to be rather inactive (compare red dashed and solid lines in Fig. 11).

Finally in the layer below 1km tracer concentrations are fairly low throughout most of the period in all four subregions. At the end of 02 July moderate increases are seen in the two westernmost domains. These appear to be related to the showers discussed in the context of Figs. 5–8 in the previous section. In the afternoon and evening hours of 03 July a more marked mixing event occurs in the two western domains with concentrations peaking around 18 UTC. Reflections of these events can be seen in the upper layers as well. Turbulent diffusion practically plays no role in the mass increase in the lower layer. The event on 03 July contributes most to the large-scale increase seen in Fig. 9c. This discussion illustrates the influence of localized intense mixing events on tracer concentrations in the PBL, which will largely be missed out on by models that parameterize moist convection.

## 6 Summary and conclusions

Recent observational and modelling work has revealed significant concentrations of biomass burning aerosol reaching SWA in the PBL and contributing to a deterioration of air quality there (Brito et al., 2018; Menut et al., 2018; Haslett et al., 2019). It has been suggested that this plume stems from the extensive fires in Central Africa during the WAM season. Here we investigated potential transport pathways of the aerosol. While previous studies discussed subsidence to the west of the African continent to be an important mechanism, here we identify – to the best of our knowledge – for the first time that downward cloud venting is one of the processes by which biomass burning aerosol from mid-tropospheric layers is mixed into the PBL over the Gulf of Guinea.

This study heavily relied on high-resolution simulations using the COSMO-ART model for July 2016. COSMO-ART enables us to simulate both meteorological fields and CO distributions over a domain including SWA, Central Africa and the adjacent tropical Atlantic. The simulated wind speed and direction are broadly in agreement with the ERA-Interim reanalysis, although COSMO revealed a somewhat stronger midlevel export from Central Africa and faster monsoon flow. Regarding cloud cover, COSMO-ART reproduces areas of

maximum and minimum clouds over SWA but overestimates it over the Gulf of Guinea. The spatial distribution of CO is used as a tracer to detect the biomass burning plume. Compared to observations, the simulated CO concentration capture the main spatial patterns, but the Central African biomass burning plumes appears to be overestimated.

For a particular case study (02 July 2016), we conducted simulations for realistic conditions and for idealized experiments, with a passive tracer initially restricted entirely to the 2–4km layer. The main results are schematically illustrated in Fig. 12, showing an ensemble of clouds with different vertical extents. The biomass burning aerosol is first transported out to the tropical Atlantic with a strong easterly flow at midlevels (yellow arrow in Fig. 12). Its base is often well above the usually shallow oceanic PBL (sketched to reach 800m in Fig.

12). Both turbulent diffusion and cloud-induced mixing cause a vertical transport to below 1km with the latter contributing more than 2/3 over most areas. Individual cloud-induced mixing events can be detected that are associated with deeper clouds, precipitation and downdrafts leading to surface cold pools. Concentrations of the biomass burning plume aerosol below 1km reach about 15% of the initial mass in the 2–4km layer after two days in our tracer experiments. Details of the mixing depend crucially on cloud depth and precipitation intensity

(as indicated by solid and stippled arrows in Fig. 12). Once in the lower layers biomass burning can be carried northward with the southerly or southwesterly monsoon winds (indicated by yellow arrows in Fig. 12). It is conceivable (but not shown here) that the strong shear between near-surface southerlies and midlevel easterlies helps tilting convective clouds when they form and thereby increases evaporation at cloud edges, downdraft formation, and mixing. In addition, we found a meridional gradient in the effectiveness of downward transport

irrespective of the actual sources of biomass burning aerosol. The largest PBL input occur over the warm waters of the southern hemisphere (around 7°S) with a marked decrease towards the equatorial cold tongue. Towards the north, SST increase again but mixing efficiency does not reach the same levels as in the southern hemisphere, possibly due to differences in vertical stability.

This study is largely based on a case study to illustrate the potential importance of downward cloud venting over

the Gulf of Guinea. More such investigations are needed based on longer simulations and other models to get more robust statistics. Moreover, the tracer experiment we presented here was performed for an inert tracer such as CO with no sedimentation and no deposition. The Central African biomass burning plume contains large amounts of aerosols, which do sediment and can be washed out by rainfall into the ocean. The magnitude of this, however, remains an open question that needs to be addressed in future studies.


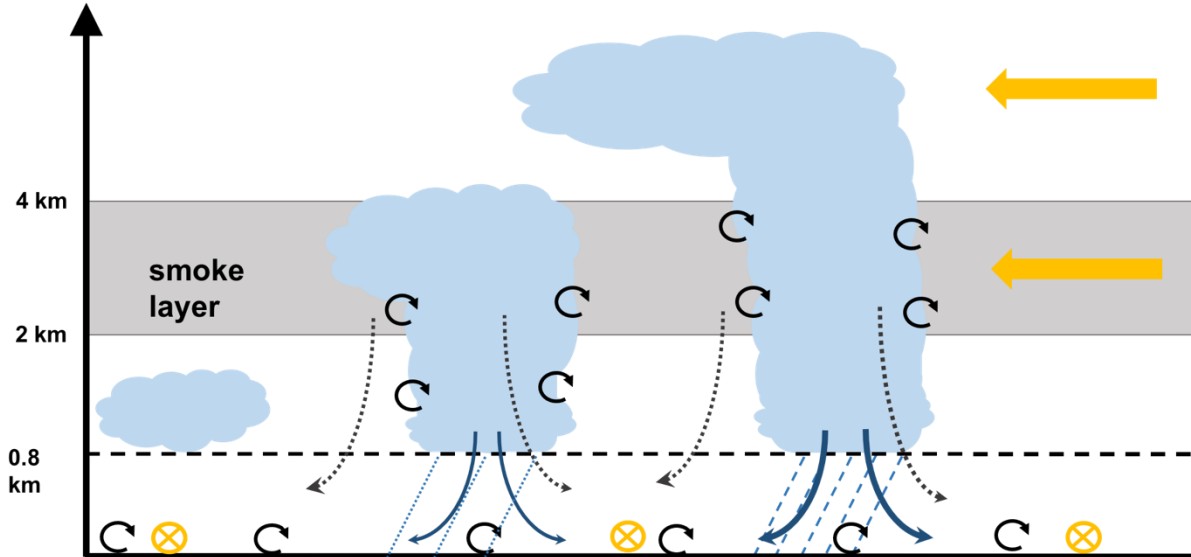

*Figure 12: Sketch of the vertical exchange processes that transport aerosol from the Central African smoke layer into the PBL over the tropical Atlantic. Shown is an Instantaneous ensemble of different cloud types with different vertical extents over the equatorial Atlantic. Blue solid (black stippled) arrows indicate downdrafts from below cloud base (cloud edges) of different strengths. Easterly flow at mid-levels is indicated by yellow arrows while the southerly monsoon flow at low levels is marked by yellow circles with crosses. Turbulent diffusion is indicated by black swirls.*

**Authors Contributions.** D.A., B.V., K.O. and V.Y. conceived and designed the study. D.A., B.V. and H.V. developed the model codes and carried out the simulations. D.A., P.K., B.V., and K.O. contributed to the literature, data analysis/interpretation and manuscript writing. P.K., B.V., S.S., E.T.N., and V.Y. contributed to the manuscript revision.

**Competing interests**. The authors declare that they have no conflict of interest.

**Acknowledgments**. The research leading to these results has received funding from West African Climate Science Service for Climate Change and Adapted Land Use (WASCAL) and got support from European Union 7th Framework Programme (FP7/2007-2013) under Grant Agreement no. 603502 (EU project DACCIWA: Dynamics-aerosol-chemistry-cloud interactions in West Africa). The first author would like to thank Aerosols, Trace Gases and Climate Processes, Institute of Meteorology and Climate Research –Department Troposphere Research (IMK-TRO) research group for hosting her during one year at Karlsruhe institute of Technology (KIT) and for their valuable contribution to the paper. We acknowledge the constructive comments from two anonymous reviewers that have helped significantly to improve the manuscript.

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
