# Peer review of "Downward cloud venting of the Central African biomass burning plume during the West Africa summer monsoon"

_Atmospheric Chemistry and Physics, 2019_

## Referee Comment (RC1) · Anonymous Referee #1 · 19 Aug 2019

General Comments

The manuscript has an important goal–investigate the hypothesis that "cloud venting plays a considerable role in the downward mixing of biomass burning aerosol" (line 74). Achieving that goal would be of great interest to everyone in the field because so little is known about it. Unfortunately, the manuscript does a very poor job of investigating the topic. Many figures are shown and discussed, but none directly address the cloud venting issue. There is no systematic laying out of evidence to support the hypothesis. Nonetheless, the authors make unsupported speculations about how cloud venting could be a factor in what is being shown. Sad to say, but after finishing the manuscript

I felt that I had learned nothing about cloud venting.

Major Issues

1. The Introduction contains no information about previous studies of downward transport by convection (your topic). If any exist, they should be described. Your work should pick up where the previous studies left off.

2. Line 150–It is unclear how the case study of 02-03 July was conducted. Did you use the COSMO simulated meteorology, but instead of the MOZART chemistry, you specified the simplified CO profile? The details of the case study methodology must be better described. ***This description will be critical to the success of the manuscript. Readers must know what you did before they can assess what the results show.

3. State the reason why you chose 02-03 July for the case study. Was it typical of all July days or was it picked for a particular reason?

4. Line 221–Observed vs. simulated CO in Fig. 4. There are some major differences in the two fields. You say that features are "reasonably captured". That is a very generous assessment that I believe should be 'toned down'.

5. Line 240-1–As mentioned in the General Comments, I maintain that you do not 'especially focus on the role of convective clouds'. That is my fundamental problem with the manuscript; the results seldom address the hypothesis.

6. Line 275–"conspicuous north-south orientation" of cells. I don't see this.

7. Section 4.2–The oceanic area should be the focus of this section. While it is good to have simulated/observed agreement over land, it is especially important that this area be well simulated because you later will show cross sections and area averages in this region. So…a change in focus is needed in this section.

8. Figure 6– Panel c) contains simulated precipitation rate at 1800 UTC while panels a) and b) are for 1200 UTC. Why the time change? The text says nothing about convective

evolution between the two times. Also, does panel b) comprise the same area as the other panels? There are no lat/long indications on b).

9. Figure 7 does not cover the same domain as Figs. 5 or 6. This makes a comparison of features very difficult. I suggest you place a dashed line at 4 N as done in earlier figures. Please outline the area in Fig. 7 that corresponds to the area in Fig. 6.

Also, Fig. 8 will show that the level of maximum CO is $\sim$ 2000 m. That would seem a better choice for Fig. 7 than 2900 m. Or. . ...you could add 2000 m as a third panel to Fig. 7.

10. Line 311–There is no justification for this transport statement.

11. Line 313–I do not see a CO feature at 2900 m that is "west and north of the main plume". Also, what is your rationale for stating that this could be due to downward mixing?

12. Line 331–How does Fig. 8 indicate that the biomass plume is advected in a westerly direction?

13. Line 341 and Fig. 8d (cloud liquid water)—How does this panel indicate positive and negative vertical motions?

14. Line 345–What do you mean by "less evidence of convective mixing"?

15. Line 361, simulations initialized—Is this truly a new and different model run initialized on 2 July? I had assumed that you were using the meteorology for 2-3 July from the simulation begun on 25 June (the run you have been describing up to this point). However, now you have used the simple CO profile, not the MOZART-derived CO. I am confused. You must explain what you are doing. This is the same issue raised in question 2 above.

16. Figure 9–You show the layers below 1 km, between 2-4 km, and the sum "between the two". Why did you not include the layer between 1-2 km?

17. Line 399 + and Fig. 10–I do not understand the purpose of relating SST to % mass. How does this relate to cloud venting? You must explain the relevance of this figure to your hypothesis.

Minor Issues

1. Describing locations by city names should be avoided. Most readers will have insufficient geographic knowledge of these city locations. Either provide a map showing all cities that are mentioned or completely avoid the use of city names.

2. Line 146–What are the horizontal and vertical grid spacings of the ICON and MOZART data that are used for your ICs and BCs? State these in the text.

3. Line 151–Is your model configured with one-way or two-way interactions between D1 and D2? State this in the text.

4. Line 165–I assume that D1 and D2 are being run concurrently. However, line 165 ("over D1") gives me doubt. Please re-phrase to make this clear.

5. Figure 2 should be in color like the other figures.

6. Figure 3b–The abrupt gradient in MODIS-derived cloud cover at 4.5 N is very suspicious looking. Could this be a data problem, or is it real?

7. Figure 7 caption–Do you mean that all the cooler regions west of 3 W over the water are cold pools related to convective cells? If only certain regions represent cold pools, those should be denoted by arrows. Also, it would strengthen your argument that these are convectively related cold pools by referring to Fig. 6c which shows that there was plenty of simulated rainfall in the area (at least 6h later at 1800 UTC).

8. Line 347 (Fig. 8f)–The panel label for f) says 1 deg W, not 1 deg E.

9. Figure 9–Dates on the x-axis should be labeled.

Please also note the supplement to this comment:

https://www.atmos-chem-phys-discuss.net/acp-2019-617/acp-2019-617-RC1-supplement.pdf

---

## Referee Comment (RC2) · Anonymous Referee #2 · 19 Sep 2019

This manuscript describes the simulation of meteorology and transport of CO from biomass burning in Central Africa and its arrival in the boundary layer over the Gulf of Guinea. The model did a reasonably good job with this simulation, and showed that convection over the Gulf was capable of downward transport of CO from the elevated plume into the boundary layer. The manuscript is quite well written. I have several minor comments as listed below, and once they are addressed I think the manuscript can be accepted for publication.

line 63: emission of what?

line 66: up to 30%

line 76: add reference to Dickerson et al. (1987). This was the first observational evidence of the upward transport of pollutants by deep convection.

line 84: TRACE-A

line 87: Please add another TRACE-A reference: Pickering et al. (1996) simulated observed convective transport of biomass burning emissions over Brazil in TRACE-A and their downwind transport over the Atlantic.

Figure 3: need to mention again that the model gets too much cloud over ocean. Also note not enough cloud north of 8 degrees N.

line 220: Is this the monthly mean of CO from the model over each day of the month at the MOPITT overpass time? If so, then you need to say that. If it is not that, then it needs to be corrected.

line 226: How does MOPITT perform for surface CO? You need to cite some literature concerning the validation of the MOPITT product at this level version some in-situ observations. Does MOPITT CO at the surface have a low or high bias?

line 248: Are the more pronounced cooler areas over land also related to convective cells?

Figure 6: Point out again that the model produced too many convective cells over the ocean.

line 316: What about convection over land. Did it also show downward mixing?

line 339: Figure 8a and 8b are zonal cross sections

line 346: I can't see that CO in the PBL is less than at 6.1 degrees W. It is the same color.

line 347: Figure 8f says 1 degree W. The text and caption say 1 degree E. Please correct one or the other.

[Figure]

line 378: add reference to Figure 9c here

lines 395-396: west to east, i.e., from A to D

---

## Author Comment (AC1) · 15 Feb 2020

Referee #1 We thank the Reviewer for her/his time to review our manuscript and for the constructive criticism. Please find below our point-by-point answers in red, modifications of the manuscript are in blue while the original Reviewer's comments are in black.

[Figure]

General Comments The manuscript has an important goal–investigate the hypothesis that "cloud venting plays a considerable role in the downward mixing of biomass burning aerosol" (line 74). Achieving that goal would be of great interest to everyone in the field because so little is known about it. Unfortunately, the manuscript does a very poor job of investigating the topic. Many figures are shown and discussed, but none directly address the cloud venting issue. There is no systematic laying out of evidence to support the hypothesis. Nonetheless, the authors make unsupported speculations about how cloud venting could be a factor in what is being shown. Sad to say, but after finishing the manuscript I felt that I had learned nothing about cloud venting.

We agree that this aspect was not as clear as it should have been in the first version of this paper. We have now substantially rewritten the section on the tracer experiment to much better support the argument. Tracer experiments with and without turbulent diffusion help to clearly isolate the cloud effect. We hope that this way the evidence is clear and convincing.

Major Issues

1. The Introduction contains no information about previous studies of downward transport by convection (your topic). If any exist, they should be described. Your work should pick up where the previous studies left off.

We agree that this aspect was in fact underrepresented in the Introduction. The following paragraph on downward transport is now included from line 101 onwards:

In contrast, rather few studies investigated the downward transport of elevated pollution through convective clouds. For the marine PBL, aerosol particles from the free troposphere have been identified to serve as cloud condensation nuclei in stratiform clouds with cloud entrainment contributing up to 20% of the aerosol mass (Raes, 1995; Katoshevski et al. 1999). Over land, most studies concentrated on the Amazon rainforest. Based on campaign data during the wet season, Betts et al. (2002) showed that convective downdrafts rapidly transport air with high ozone down to the surface from

around 800 hPa, suggesting a significant role of this process for the photochemistry of the PBL and surface ozone deposition. Gerken et al. (2016) even found evidence for transport of ozone-rich air from the mid-troposphere to the surface, enhancing the volume mixing ratio in the boundary layer by as much as 25 ppbv on the regional scale, while Wang et al. (2016) demonstrated the injection of high concentrations of small aerosol particles into the PBL by strong convective downdrafts. In more general terms, Jonker et al. (2008) proposed a refined view of mass transport by cumulus convection relevant for the dispersion of aerosol. According to them, the descending motion near the cloud environment is significant and rather different than in a distant cloud environment, which is characterized by more uniform and quiescent dry descending motion.

2. Line 150–It is unclear how the case study of 02-03 July was conducted. Did you use the COSMO simulated meteorology, but instead of the MOZART chemistry, you specified the simplified CO profile? The details of the case study methodology must be better described. ***This description will be critical to the success of the manuscript. Readers must know what you did before they can assess what the results show.

We acknowledge that this description was not as detailed as it should have been. The following information has now been added to the modelling section 2.2 in line 195:

Both domains, D1 and D2, were run with the parametrization for deep convection switched off and using the two-moment microphysics scheme (Seifert and Beheng, 2006). Over D1, the modelled period ranges from 25 June–31 July 2016 with the meteorological state being re-initialized every day at 00 UTC. ICON operational forecasts at 13 km grid spacing with 90 vertical levels are used as meteorological initial and boundary conditions and MOZART chemistry with a grid mesh of 280 km x 213 km and 56 vertical levels for the pollutant initial and boundary data. Cloud condensation nuclei are prescribed with a constant aerosol number concentration of 1700 cm-3. The purpose of the D1 simulation is to compare the model output and observations for monthly mean conditions, i.e., for July 2016, after a six-day spin-up.

3. State the reason why you chose 02-03 July for the case study. Was it typical of all July days or was it picked for a particular reason?

Good point. We added the following text to section 2:

The period 02–03 July 2016 was chosen because it falls into the post onset phase of the monsoon, characterizing an undisturbed monsoon condition, and is thus favorable for process studies (Knippertz et al., 2017; Deetz et al., 2018).

4. Line 221–Observed vs. simulated CO in Fig. 4. There are some major differences in the two fields. You say that features are "reasonably captured". That is a very generous assessment that I believe should be 'toned down'.

This sentence was modified according to your comment (line 264):

Overall the spatial patterns of CO concentration are captured by the model with some discrepancies.

5. Line 240-1–As mentioned in the General Comments, I maintain that you do not 'especially focus on the role of convective clouds'. That is my fundamental problem with the manuscript; the results seldom address the hypothesis.

We have now created a new section 5 named "Downward cloud venting" that is fully dedicated to this topic. We have run additional tracer experiments with suppressed turbulent diffusion in case of the tracer to better isolate the effects of downward cloud venting. Turbulent diffusion is accounted for for all other variables (wind, temperature, humidity, hydrometeors). We also created a new figure in the summary section to better illustrate the mechanisms at play (see Fig. 12 in the manuscript). Here is what we wrote in this new section at line 424 in the manuscript:

In general, processes that can support the transport of biomass burning aerosols from free-tropospheric layers into the PBL include: (i) large-scale subsidence and thus vertical advection (Katoshevski et al., 1999), (ii) turbulent mixing into the marine PBL, and (iii) vertical transport by convective clouds. With respect to point (i) we can state

that the cross sections in Fig. 8 do not show clear indications of a systematic sinking of the biomass burning plume suggesting that for the situation presented in section 4 synoptic-scale subsidence is not a leading factor. To investigate the relative importance of processes (ii) and (iii), we designed an idealized tracer experiment. For the simulations starting at 02 July 2016 at 00 UTC initial profiles of a tracer were prescribed within the domains D1 and D2. The idealized tracer has a concentration of 1 ppmv between 2 and 4 km and is zero elsewhere. Chemical reactions as well as dry deposition are neglected in order to isolate effects of transport only. At the lateral boundaries the tracer concentrations were held constant at the initial profile such that only mixing within the domain can change tracer concentrations. Two sets of simulation were done: one with and one without turbulent diffusion. The idea behind this is to separate this effect from that of downward cloud venting. The simulations were carried out for a period of two days (2–3 July 2016).

6. Line 275–"conspicuous north-south orientation" of cells. I don't see this.

This statement has now been updated at line 320 in the manuscript.

The largest and most intense convective systems are simulated over the ocean with a pronounced north–south elongation along the southwesterly monsoon flow. These were persistent throughout the day (not shown).

7. Section 4.2–The oceanic area should be the focus of this section. While it is good to have simulated/observed agreement over land, it is especially important that this area be well simulated because you later will show cross sections and area averages in this region. So . . .a change in focus is needed in this section. We modified this section slightly to emphasis the ocean a little bit. However, in the discussion of Fig. 9 we also have a part on the land situation and therefore think that the description we provide here is necessary.

8. Figure 6– Panel c) contains simulated precipitation rate at 1800 UTC while panels a) and b) are for 1200 UTC. Why the time change? The text says nothing about convective

C2 evolution between the two times. Also, does panel b) comprise the same area as the other panels? There are no lat/long indications on b).

The reason for this choice is to illustrate the appearance of convective cells prior to the rainfall. Concerning, panel b) it corresponds to the same area and indications of latitude and longitude have now been added. The figure below presents the evolution of the convective cells between 13 UTC and 18 UTC. They appeared over the Gulf of Guinea around 7 UTC from were persistent throughout the day.

Temporal evolution of convective cells from SEVIRI cloud visible image EUMETSAT from 13 UTC to 18 UTC (from http://nascube.univ-lille1.fr).

9. Figure 7 does not cover the same domain as Figs. 5 or 6. This makes a comparison of features very difficult. I suggest you place a dashed line at 4 N as done in earlier figures. Please outline the area in Fig. 7 that corresponds to the area in Fig. 6. Also, Fig. 8 will show that the level of maximum CO is 2000 m. That would seem a better choice for Fig. 7 than 2900 m. Or :you could add 2000 m as a third panel to Fig. 7. We agree with your suggestions. Fig. 7 has now been replaced by the figure below. The text was modified in the following way (line 351):

Concentrations over the nested domain D2 at 500m (Fig 7c) are moderated with traces of higher CO concentrations over the Gulf of Guinea, some smaller elongated plumes (e.g., from Abidjan and Accra), and much elevated levels downstream of Lake Volta. As concentrations above ground level are shown in Fig. 7c, the elevated values over the Atakora chain at the border of Ghana with Togo are at least party related to the fact that higher ground is closer in the vertical to the main midlevel pollution plume from Central Africa.

10. Line 311–There is no justification for this transport statement.

Areas where concentrations are low at 500 m and elevated at 2000 m cannot be dominated by local sources at the surface. Based on previous studies, the bulk of biomass

burning plume is located between 2 and 4 km of altitude. Furthermore, our simulations with and without biomass burning (not shown) confirm that high concentration found at 2000 m results from biomass burning emissions. We there argue that this statement is justified and did not change it.

11. Line 313–I do not see a CO feature at 2900 m that is "west and north of the main plume". Also, what is your rationale for stating that this could be due to downward mixing?

We agree that this is not easy to understand. We changed the text in the following way:

The elevated concentrations at 500m over the ocean to the west and north of the main plume at 2000m suggest downward mixing into the PBL from aloft.

12. Line 331–How does Fig. 8 indicate that the biomass plume is advected in a westerly direction?

You are right that this is an overinterpretation. We changed the sentence in the following way:

There is a clear band of high CO concentrations of up to 400 ppbv, mostly between 1 and 3.5km over D2, which is the signature of the long-range transport of the biomass burning plume from Central Africa (Mari et al., 2008; Zuidema et al., 2016), possibly affected by larger-scale subsidence.

13. Line 341 and Fig. 8d (cloud liquid water)—How does this panel indicate positive and negative vertical motions?

We agree that this was misleading. The respective section now reads:

Figures 8c and d show meridional-vertical cross-sections of, respectively, CO concentration and specific cloud liquid water content along 6°W, close to where convective activity is seen in Fig. 8a. Areas of high cloud liquid water are collocated with minima in CO, supporting the idea of cloud-induced transport and mixing. The most prominent of such areas is located around 4.3°N, where significant amounts of cloud water stretch from below 500m to almost the top of the biomass burning plume, leading to an substantial erosion.

14. Line 345–What do you mean by "less evidence of convective mixing"?

What we mean here is that, Fig. 8e does not show the sharp gaps in the pollution plume as evident from Figs. 8 a and b. This is now explicitly mentioned in the text:

At 4°W (Fig. 8e) there are no pronounced gaps in the pollution plume, suggesting less convective mixing at this time than at 6°W but concentrations at low levels are not much different.

15. Line 361, simulations initialized—Is this truly a new and different model run initialized on 2 July? I had assumed that you were using the meteorology for 2-3 July from the simulation begun on 25 June (the run you have been describing up to this point). However, now you have used the simple CO profile, not the MOZART-derived CO. I am confused. You must explain what you are doing. This is the same issue raised in question 2 above.

No, these are in fact new simulations as now clearly explained in the text.

16. Figure 9–You show the layers below 1 km, between 2-4 km, and the sum "between the two". Why did you not include the layer between 1-2 km?

We agree that this should be included. The mass calculation between 1-2 km have been now added in Figures 9 and 11 corresponding to the SWA and regional domains, respectively.

17. Line 399 + and Fig. 10–I do not understand the purpose of relating SST to % mass. How does this relate to cloud venting? You must explain the relevance of this figure to your hypothesis.

This Figure (Fig. 10) investigates the influence of the SST on the transported mass of

the tracer to below 1km after two days of integration with respect to latitude. This is related to the cloud venting, because in the discussion of Fig. 9 we show that this is the dominant process in the vertical transport. Here we see that SST modulates the mass transported into the PBL. Based on the findings from Fig. 9 we assume that this due to clouds. We rewrote this entire section to explain that better.

Minor Issues 1. Describing locations by city names should be avoided. Most readers will have insufficient geographic knowledge of these city locations. Either provide a map showing all cities that are mentioned or completely avoid the use of city names.

Fig. 1 has been replaced by figure with city names.

2. Line 146–What are the horizontal and vertical grid spacings of the ICON and MOZART data that are used for your ICs and BCs? State these in the text.

This has now been added to the text at line 198 as follows:

ICON operational forecasts at 13 km grid spacing with 90 vertical levels are used as meteorological initial and boundary conditions and MOZART chemistry with a grid mesh of 280 km x 213 km and 56 vertical levels for the pollutant initial and boundary data

3. Line 151–Is your model configured with one-way or two-way interactions between D1 and D2? State this in the text. This has now been updated.

4. Line 165–I assume that D1 and D2 are being run concurrently. However, line 165 ("over D1") gives me doubt. Please re-phrase to make this clear.

D1 was run first and then D2 is nested into the coarse domain D1. This was re-phrased as follow in the manuscript at line 204:

we analyze a particular case study on 02–03 July 2016 simulated over D2 using the outputs of D1 for both meteorological and chemical initial and boundary conditions. The TMMS was combined with the prognostic aerosol, this way accounting for aerosol

direct and indirect interactions.

5. Figure 2 should be in color like the other figures.

This figure has now been updated.

6. Figure 3b–The abrupt gradient in MODIS-derived cloud cover at 4.5 N is very suspi-cious looking. Could this be a data problem, or is it real?

The data have been checked and appear to be correct. One possible explanation we mention in the text is areas of coastal upwelling that may locally modify cloud cover.

7. Figure 7 caption–Do you mean that all the cooler regions west of 3 W over the water are cold pools related to convective cells? If only certain regions represent cold pools, those should be denoted by arrows. Also, it would strengthen your argument that these are convectively related cold pools by referring to Fig. 6c which shows that there was plenty of simulated rainfall in the area (at least 6h later at 1800 UTC).

Figure 7 has been updated now and the cool pools are more visible over the D2 do-main. The following text at line 368 in the manuscript has been added:

Zooming in on domain D2 (Fig. 7d), concentrations at 2000m are generally much higher than at 500m (Fig. 7c), in particular over the coastal zone. Strikingly some marked "holes" are evident that correspond to areas of cold pools associated with convective cells (see Figs. 5 and 6c), suggesting that in these areas clouds support downward mixing.

8. Line 347 (Fig. 8f)–The panel label for f) says 1 deg W, not 1 deg E.

The caption of Fig. 8f has been corrected. It is 1°E.

9. Figure 9–Dates on the x-axis should be labeled.

This has been added now.

References Betts, A. K., Gatti, L. V., Cordova, A. M., Silva Dias, M. A. F. and

Fuentes, J. D.: Transport of ozone to the surface by convective downdrafts at night, J. Geophys. Res. D Atmos., 107(20), 1–7, doi:10.1029/2000JD000158, 2002. Eastman, R., Warren, S. G. and Hahn, C. J.: Variations in cloud cover and cloud types over the Ocean from surface observations, 1954-2008, J. Clim., 24(22), 5914–5934, doi:10.1175/2011JCLI3972.1, 2011. Gerken, T., Wei, D., Chase, R. J., Fuentes, J. D., Schumacher, C., Machado, L. A. T., Andreoli, R. V., Chamecki, M., Ferreira de Souza, R. A., Freire, L. S., Jardine, A. B., Manzi, A. O., Nascimento dos Santos, R. M., von Randow, C., dos Santos Costa, P., Stoy, P. C., Tóta, J. and Trowbridge, A. M.: Downward transport of ozone rich air and implications for atmospheric chemistry in the Amazon rainforest, Atmos. Environ., 124(November), 64–76, doi:10.1016/j.atmosenv.2015.11.014, 2016. Jonker, H. J. J., Heus, T. and Sullivan, P. P.: A refined view of vertical mass transport by cumulus convection, Geophys. Res. Lett., 35(7), 1–5, doi:10.1029/2007GL032606, 2008. Katoshevski, D., Nenes, A. and Seinfeld, J. H.: A study of processes that govern the maintenance of aerosols in the marine boundary layer, J. Aerosol Sci., 30(4), 503–532, doi:10.1016/S0021-8502(98)00740-X, 1999. Kaufman, Y. J., Koren, I., Remer, L. A., Rosenfeld, D. and Rudich, Y.: The effect of smoke, dust, and pollution aerosol on shallow cloud development over the Atlantic Ocean, Proc. Natl. Acad. Sci., 102(32), 11207–11212, doi:10.1073/pnas.0505191102, 2005. Raes, F.: Entrainment of free tropospheric aerosol as a regulation mechanism for cloud dondensation nuclei in the remote marine boundary layer, , 100, 2893–2903, 1995. Seifert, A. and Beheng, K. D.: A two-moment cloud microphysics parameterization for mixed-phase clouds. Part 1: Model description, Meteorol. Atmos. Phys., 92(1–2), 45–66, doi:10.1007/s00703-005-0112-4, 2006.
* * *
[Figure: satellite imagery panels labeled 13 UTC, 14 UTC, 15 UTC, 16 UTC, 17 UTC, 18 UTC]

**Fig. 1.**

[Figure]

**Fig. 2.**

---

## Author Comment (AC2) · 15 Feb 2020

Referee #2 We thank the Reviewer for her/his time to review our manuscript and for the constructive criticism. Please find below our point-by-point answers in red, modifications of the manuscript are in blue while the original Reviewer's comments are in black.

[Figure]

line 63: emission of what? This sentence is now in line 64 and reads Biomass burning is an important source of aerosols and trace gases, with an estimated combined emission of several thousand Tg a-1 for tropical areas.

line 66: up to 30% This has now been corrected (line 68).

line 76: add reference to Dickerson et al. (1987). This was the first observational evidence of the upward transport of pollutants by deep convection. This reference has now been added in the manuscript in line 83: The hypothesis we investigate in this paper is that clouds play a considerable role in the downward mixing of biomass burning aerosol from the elevated plume. Most previous studies have focused on cloud-induced upward transport of aerosols and chemical species from close to their sources in the PBL to the free troposphere (e.g., Dickerson et al., 1987; Ching et al., 1988; Cotton et al., 1995

line 84: TRACE-A This has been corrected at line 92 in the manuscript.

line 87: Please add another TRACE-A reference: Pickering et al. (1996) simulated observed convective transport of biomass burning emissions over Brazil in TRACE-A and their downwind transport over the Atlantic. Pickering et al. (1996) showed an upward transport of CO mixing ratios, NOx and hydrocarbons by convective clouds during the Brazilian phase of TRACE-A experiment". This information has now been added in line 96.

Figure 3: need to mention again that the model gets too much cloud over ocean. Also note not enough cloud north of 8 degrees N. This information has now been updated in line 253 in the manuscript: Towards the Sahel, to the north of 8° N, cloud fraction decreases in COSMO-ART but much less so in MODIS, which only shows a prominent minimum over central Ivory Coast. Over the Gulf of Guinea, cloud cover is clearly overestimated by the model...

line 220: Is this the monthly mean of CO from the model over each day of the month

at the MOPITT overpass time? If so, then you need to say that. If it is not that, then it needs to be corrected. This has been modified in the manuscript at line 262: A gridded monthly mean of CO from MOPITT is computed using the daily mean CO retrieved for the 1000–900 hPa layer.

line 226: How does MOPITT perform for surface CO? You need to cite some literature concerning the validation of the MOPITT product at this level version some in-situ observations. Does MOPITT CO at the surface have a low or high bias? The following text has been added in line 141 in the manuscript: MOPITT data have been shown to distinguish CO pollution from large cities and urban areas from background pollution using only thermal infrared information (Clerbaux et al., 2008) and to perform even better using a combination of thermal infrared and solar radiation in the PBL (Buchwitz et al., 2007; Turquety et al., 2008). Kar et al. (2008) highlighted that retrievals in the lower troposphere over continental areas provide reasonable information on surface emissions of CO, although the measurements suffer from strong thermal contrasts. According to Buchholz et al. (2017), MOPITT measurements overestimate relative to ground-based remote sensing Fourier transform infrared spectrometer with a bias of less than 10% evaluated over 14 stations.

line 248: Are the more pronounced cooler areas over land also related to convective cells? Yes. We added and the adjacent land areas, particularly over southern Ivory Coast in line 293.

Figure 6: Point out again that the model produced too many convective cells over the ocean. We agree that this point should be highlighted. It has been now indicated in the text in line 320: The largest and most intense convective systems are simulated over the ocean with a pronounced north–south elongation along the southwesterly monsoon flow. These were persistent throughout the day (not shown).

line 316: What about convection over land. Did it also show downward mixing? Yes, it does. This has been addressed and further discussed in the manuscript from line 475

onwards. Finally over land, i.e., between 5 and 10°N, the vertical exchange maximizes leading to a reduction in the 2–4 km layer down to almost 50% (Fig. 9d). Consistently, tracer mass in the intermediate layer increases more strongly up to well over 40%, while tracer mass below 1 km reaches 23%. A clear diurnal cycle is evident, particularly in the lower layer, with vertical mixing mostly occurring in the early afternoon when the PBL is deepest. The suppressing of turbulent diffusion reduces the tracer mass by 20% with some evidence of a diurnal cycle in the differences evident from Fig. 9d. As expected, dry mixing is more important in the early afternoon, while cloud-induced mixing peaks later. The important role of clouds in vertical mixing over land is consistent with the large cloud cover shown in Fig. 3. The diurnal cycle is also evident at 1–2km, where switching-off turbulent diffusion leads to a net increase in this layer during the afternoon.

line 339: Figure 8a and 8b are zonal cross sections This has now been corrected.

line 346: I can't see that CO in the PBL is less than at 6.1 degrees W. It is the same color. You are right. We reworded this part: At 4°W (Fig. 8e) there are no pronounced gaps in the pollution plume, suggesting less convective mixing at this time than at 6°W but concentrations at low levels are not much different. There is even a slight increase northwards that may come from turbulent mixing or zonal advection into the section.

line 347: Figure 8f says 1-degree W. The text and caption say 1-degree E. Please correct one or the other. The caption of Fig. 8f has now been corrected. It is 1°E.

line 378: add reference to Figure 9c here This figure has been updated and the discussion of the results of the tracer experiments was extended.

lines 395-396: west to east, i.e., from A to D This sentence has been corrected.

References Barthe, C., Mari, C., Chaboureau, J. P., Tulet, P., Roux, F. and Pinty, J. P.: Numerical study of tracers transport by a mesoscale convective system over West Africa, Ann. Geophys., 29(5), 731–747, doi:10.5194/angeo-29-731-2011, 2011. Buchholz, R. R., Deeter, M. N., Worden, H. M., Gille, J., Edwards, D. P., Hannigan, J. W., Jones, N. B., Paton-Walsh, C., Griffith, D. W. T., Smale, D., Robinson, J., Strong, K., Conway, S., Sussmann, R., Hase, F., Blumenstock, T., Mahieu, E. and Langerock, B.: Validation of MOPITT carbon monoxide using ground-based Fourier transform infrared spectrometer data from NDACC, Atmos. Meas. Tech., 10(5), 1927–1956, doi:10.5194/amt-10-1927-2017, 2017. Buchwitz, M., Khlystova, I., Bovensmann, H. and Burrows, J. P.: and Physics Three years of global carbon monoxide from SCIA-MACHY : comparison with MOPITT and first results related to the detection of enhanced CO over cities, , 6, 2399–2411, 2007. Clerbaux, C., George, M., Turquety, S., Walker, K. A., Barret, B., Bernath, P., Boone, C., Borsdorff, T., Cammas, J. P., Catoire, V., Coffey, M., Coheur, P.-F., Deeter, M., De Mazière, M., Drummond, J., Duchatelet, P., Dupuy, E., de Zafra, R., Eddounia, F., Edwards, D. P., Emmons, L., Funke, B., Gille, J., Griffith, D. W. T., Hannigan, J., Hase, F., Höpfner, M., Jones, N., Kagawa, A., Kasai, Y., Kramer, I., Le Flochmoën, E., Livesey, N. J., López-Puertas, M., Luo, M., Mahieu, E., Murtagh, D., Nédélec, P., Pazmino, A., Pumphrey, H., Ricaud, P., Rinsland, C. P., Robert, C., Schneider, M., Senten, C., Stiller, G., Strandberg, A., Strong, K., Sussmann, R., Thouret, V., Urban, J. and Wiacek, A.: CO measurements from the ACE-FTS satellite instrument: data analysis and validation using ground-based, airborne and spaceborne observations, Atmos. Chem. Phys., 8(9), 2569–2594, doi:10.5194/acp-8-2569-2008, 2008. Das, S., Harshvardhan, H., Bian, H., Chin, M., Curci, G., Protonotariou, A. P., Mielonen, T., Zhang, K., Wang, H. and Liu, X.: Biomass burning aerosol transport and vertical distribution over the South African-Atlantic region, J. Geophys. Res., 122(12), 6391–6415, doi:10.1002/2016JD026421, 2017. Gerken, T., Wei, D., Chase, R. J., Fuentes, J. D., Schumacher, C., Machado, L. A. T., Andreoli, R. V., Chamecki, M., Ferreira de Souza, R. A., Freire, L. S., Jardine, A. B., Manzi, A. O., Nascimento dos Santos, R. M., von Randow, C., dos Santos Costa, P., Stoy, P. C., Tóta, J. and Trowbridge, A. M.: Downward transport of ozone rich air and implications for atmospheric chemistry in the Amazon rainforest, Atmos. Environ., 124(November), 64–76, doi:10.1016/j.atmosenv.2015.11.014, 2016. Kar, J.,

Jones, D. B. A., Drummond, J. R., Attie, J. L., Liu, J., Zou, J. and Nichitiu, F.: Measurement of low-altitude CO over the Indian subcontinent by MOPITT, , 113, 1–13, doi:10.1029/2007JD009362, 2008. Turquety, S., Clerbaux, C., Law, K., Coheur, P. F., Cozic, A., Szopa, S., Hauglustaine, D. A., Hadji-Lazaro, J., Gloudemans, A. M. S., Schrijver, H., Boone, C. D., Bernath, P. F. and Edwards, D. P.: CO emission and export from Asia: An analysis combining complementary satellite measurements (MOPITT, SCIAMACHY and ACE-FTS) with global modeling, Atmos. Chem. Phys., 8(17), 5187–5204, doi:10.5194/acp-8-5187-2008, 2008.

---

## Referee Report (RR1)

Review of the Revised Manuscript

"Downward cloud venting of the Central African biomass

Burning plume during the West Africa summer monsoon"

Dajuma et al.

acp-2019-617

The authors have made major revisions to the original manuscript and have acted on all of the reviewers' comments. The manuscript is much improved. I now understand the parts of the methodology that originally were unclear. I only raise a few points that should be included in the final version. Figure 12 is a serious concern; the other points are relatively minor and only require minor wording or figure changes.

1. Fig. 2--You use different color scales at 925 and 700 hPa. I believe you could use the 700 hPa color scale (0-12 m/s) at 925 hPa. Using the same color scale would make the comparison easier for the readers.
2. Line 293—You could refer the reader to Fig. 5 to see the cold pools that you mention on line 293. You could then refer to Fig. 6c to see the convection. These reference would not require the figures to be re-numbered.
3. Line 361-3—"…main burning areas in the southern hemisphere." Are you referring to burning in Africa or South America? Since you say "long range" in line 363, I suspect you mean South America. The TRACE-A experiment proved that CO from South American can be transported to Africa where it is reinforced by African biomass burning before being transported farther east. However, when readers get to Line 379, long range seems to mean from central Africa (not nearly as long range as South America).

    If you are unsure about the source, backward trajectories using the online version of Hysplit would provide an answer. However, if Africa is indeed the source, you probably could deduce that by looking at low altitude constant pressure maps. In any event you should provide some type of proof that you know the source. No figures would be needed.
4. Line 365—"west and north of the main plume"---I am still not sure which area you are referring to because the sentence is rather vague. You could place an arrow on the figure to show it (define it in the caption). Or, you can rephrase the sentence to make the location crystal clear.
5. When I first got to line 366 and the word "suggest", I thought proof was not going to be provided. However, you do provide proof in the following paragraphs. To avoid this, you could insert a new sentence after the word "aloft", something like, "That is the subject of following paragraphs".
6. Line 408—Be specific about the source of the "indications of a slow descent". I assume you mean subsidence associated with the nearby anticyclone, but it would do no harm to mention that again.
7. Line 412—What do you mean by "some indications". Once again, please be more specific.

8. The additional information and revisions in Section 5 greatly strengthen the manuscript. Thanks.

9. ***The most important issue I have is Fig. 12. Is it a time series with later times on the right side? I assume it is, but you must clearly say so. Also, you do not show any updrafts even though the cloud is growing taller. Therefore, the figure is meteorologically impossible— growing clouds must have updrafts. I have attached a figure from an elementary meteorological text that shows what I mean. Your middle panel does not show any precipitation. Therefore, downdrafts are very unlikely—only after precipitation occurs (your right panel) can downdrafts begin. In some way your figure must conform to accepted meteorological understanding of single cell precipitating clouds. Figure 12 must be remade.

---

## Author Response (AR2)

**Review of the Revised Manuscript**
**"Downward cloud venting of the Central African biomass burning plume during the West Africa summer monsoon" by Dajuma et al. acp-2019-617**

We thank the Reviewer for her/his time to review the revised version of our manuscript. Please find below our point-by-point answers in red, modifications of the manuscript are in blue while the original Reviewer's comments are in black.

The authors have made major revisions to the original manuscript and have acted on all of the reviewers' comments. The manuscript is much improved. I now understand the parts of the methodology that originally were unclear. I only raise a few points that should be included in the final version. Figure 12 is a serious concern; the other points are relatively minor and only require minor wording or figure changes.

Thanks for your comments. Your constructive criticism has helped a lot to improve the manuscript.

1. Fig. 2--You use different color scales at 925 and 700 hPa. I believe you could use the 700 hPa color scale (0-12 m/s) at 925 hPa. Using the same color scale would make the comparison easier for the readers.

We agree with you. This figure has been updated in the manuscript.

2. Line 293—You could refer the reader to Fig. 5 to see the cold pools that you mention on line 293. You could then refer to Fig. 6c to see the convection. These references would not require the figures to be re-numbered.
This sentence now reads:

In Fig. 5 there are clear indications of cold pools related to convective cells developing over the Gulf over Guinea and the adjacent land areas, particularly over southern Ivory Coast (see Fig. 6c for precipitation).

3. Line 361-3 "… main burning areas in the southern hemisphere "Are you referring to burning in Africa or South America? Since you say "long range in line 363, I suspect you mean South America. The TRACE-A experiment proved that CO from South American can be transported to Africa where it is reinforced by African biomass burning before being transported farther east. However, when readers get to Line 379, long range seems to mean from central Africa (not nearly as long range as South America).
If you are unsure about the source, backward trajectories using the online version of Hysplit would provide an answer. However, if Africa is indeed the source, you probably could deduce that by looking at low altitude constant pressure maps. In any event you should provide some type of proof that you know the source. No figures would be needed.

We agree with you that this sentence was not clear enough. We are referring to burning in Africa specifically. The biomass burning aerosol originating from central/southern Africa burning activities, transported between 2 and 4 km is well documented (Chatfield et al., 1998; Mari et al., 2008; Das et al., 2017; Haslett et al., 2019). This sentence was modified according to your comment (line 361).
… main burning areas in southern and Central Africa.

4. Line 365—"West and north of the main plume" I am still not sure which area you are referring to because the sentence is rather vague. You could place an arrow on the figure to show it (define it in the caption). Or, you can rephrase the sentence to make the location crystal clear.

This sentence was rephrased to now read:

…west and north of the main plume at 2000 m (i.e. over the equatorial Atlantic Ocean near 15°W and arching into the Gulf of Guinea).

5. When I first got to line 366 and the word" suggest", I thought proof was not going to be _provided. However, you do provide proof in the following paragraphs. To avoid this, you could insert a new sentence after the word "aloft, something like, "That is the subject of following paragraph"

The following paragraph has been updated from line 366 onwards:

…suggest downward mixing into the PBL from aloft, which is further elucidated in the following paragraph.

6. Line 408—Be specific about the source of indications of a slow descent". I assume you mean subsidence associated with the nearby anticyclone, but it would do no harm to mention that again.

This sentence was modified according to your comment and is now in line 409:

Also here, a slow decent of the lower boundary of the plume is visible. This may come from large-scale subsidence associated with the southern branch of the Hadley cell and/or from turbulent mixing.

7. Line 412—What do you mean by "some indications" Once again, please be more specific.

This paragraph was modified according to your comment and now starts in line 413:

At 0°E (Fig. 8g) there is a distinct biomass burning plume centered at 5°S. The skewed shape of this feature suggests a relatively fast northward transport around 1000m above ground level. Individual mixing events are evident (green spikes underneath the main plume in Fig. 8g). North of the coast (marked by an arrow in Fig. 8g) there is a complicated vertical structure with local near-surface emissions, overhead advection, and vertical mixing to various degrees, particular during the daytime shown here.

8. The additional information and revisions in Section 5 greatly strengthen the manuscript. Thanks.

Thanks for your comment.

9. ***The most important issue I have is Fig. 12. Is it a time series with later times on the right side? I assume it is, but you must clearly say so. Also, you do not show any updrafts even though the cloud is growing taller. Therefore, the figure is meteorologically impossible—growing clouds must have updrafts. I have attached a figure from an elementary meteorological text that shows what I mean. Your middle panel does not show any precipitation. Therefore, downdrafts are very unlikely—only after precipitation occurs (your right panel) can downdrafts begin. In some way your figure must conform to accepted meteorological understanding of single cell precipitating clouds. Figure 12 must be remade.

There is a misunderstanding. Figure 12 is not a time series rather it but is meant to represent an instantaneous ensemble of different clouds.  The caption of Figure 12 has been updated with the following sentence:

Shown is an ensemble of different cloud types with different vertical extents over the equatorial Atlantic.

We also extended line 547 to read:

"… schematically illustrated in Fig. 12, showing an ensemble of clouds with different vertical extents."

---

## Author Response (AR3)

**Review of the Revised Manuscript**
**"Downward cloud venting of the Central African biomass burning plume during the West Africa summer monsoon" by Dajuma et al. acp-2019-617**

We thank the Editor for the time and effort to edit our manuscript in such a thoughtful way. Please find below our answer to the Editor's final comment.

I have several very minor comments remaining regarding Figure 12 that should be addressed. Once these changes have been made, I intend to accept your manuscript for publication. The concerns are related to Figure 12. Firstly, your caption does not adequately describe the various components of Figure 12, including what the different dotted and solid lines are supposed to the indicate, what the yellow arrow indicates, what the 'X' within the circle (flow into the page) actually indicates, the circular arrows and so on. Clearly while most of these could be correctly guessed, and while some of these are addressed in the text, such as the yellow arrow, your captions should include all of the symbols used within the figure. Then, I do have some concerns that in your middle panel you have downdrafts without any precipitation. Can downdrafts exist without precipitation below cloud base? Yes, they can, and one such mechanism, for example, would be the case in which all of the precipitation fully evaporates just as the downdraft exits cloud base (there are other mechanisms too). However, it is far more likely that there is still some precipitation evident below cloud base. I would therefore recommend that you indicate the presence of light precipitation, especially as you have included the presence of heavy precipitation in the right panel. If you purposefully do not want to indicate the presence of light precipitation, then it would make sense to include a statement in the paper explaining why there is no precipitation and as to which process is responsible for the downdrafts and associated vertical transport into the boundary (as there are a number of processes that could be driving these). In this way, it makes the figure more consistent.

We agree that the caption and figure need more details. Figure 12 has been updated now with light precipitation below cloud base in the middle panel. The caption of Fig. 12 now reads

Figure 12: ... Blue solid (black stippled) arrows indicate downdrafts from below cloud base (cloud edges) of different strengths. Easterly flow at mid-levels is indicated by yellow arrows while the southerly monsoon flow at low levels is marked by yellow circles with crosses. Turbulent diffusion is indicated by black swirls.